# Structure of the central *Staphylococcus aureus* AAA+ protease MecA/ClpC/ClpP
**Stavros Azinas** [1,3], **Karin Wallden**[1,3], **Panagiotis Katikaridis**[2,3], **Timo Jenne** [2], **Adrien Schahl** [1], **Axel Mogk** [2] ✉ & **Marta Carroni** [1] ✉

Bacterial AAA+ proteases are composed of a AAA+ partner (e.g., ClpC) and an associated peptidase (e.g., ClpP). They represent ATP-fuelled and self-compartmentalized proteolytic machines that are crucial for stress resistance and virulence. ClpC requires cooperation with adaptor proteins such as MecA for activation and complex formation with ClpP. Here, we present the cryo-EM structure of the MecA/ClpC/ClpP complex from the major pathogen *Staphylococcus aureus*. MecA forms a dynamic crown on top of the ClpC/ClpP complex with its substrate-binding domain positioned near the ClpC pore site, likely facilitating substrate transfer. ClpC/ClpP complex formation involves ClpC P-loops and ClpP N-terminal β-hairpins, which insert into the central ClpC threading channel and contact sites next to the ClpC ATPase center. ClpC and ClpP interactions are asymmetric and dictated by the activity states of ClpC ATPase subunits. ClpP binding increases ClpC ATPase and threading activities in a β-hairpin dependent manner, illuminating an allosteric pathway in the cooperation of ATPase and peptidase components in bacterial AAA+ proteases.

ATP-dependent protein degradation is a crucial cellular activity regulating signal transduction pathways and ensuring protein homeostasis in all organisms. The executing degradation machines are composed of AAA+ protein and peptidase modules, which either constitute separate entities (e.g., 26S proteasome, ClpX/ClpP) or are covalently linked (e.g., Lon, FtsH)[1–3].

Peptidases form self-compartmentalized barrel-like structures composed of two hexameric or heptameric rings[4]. The proteolytic active sites are hidden in the interior of the barrel, and substrate access is controlled by narrow gates, preventing unwanted proteolysis in the absence of AAA+ partners[5]. AAA+ proteins thereby control substrate specificity of AAA+ proteases and feed substrates into the proteolytic barrel in an ATP-fueled manner. ATP binding and turnover are executed by conserved AAA domains that also mediate AAA+ protein oligomerization into typically homo-hexameric rings with a central pore. Cryo-EM structures of multiple AAA+ proteins revealed conserved principles of hexamer organization and the mechanistic basis of ATPase and unfolding activity[6,7]. AAA+ complexes resemble a shallow spiral and are asymmetric, even when they are composed of identical subunits. Often, a seam subunit is most displaced from the ring axis and connects subunits at the top and bottom positions of the spiral arrangement. The ATPase ring includes active and inactive AAA subunits, which are proposed to constantly change their activity states. An active subunit has ATP bound and receives an arginine finger, a trans-acting

element crucial for ATP hydrolysis, from a neighboring subunit. ATP hydrolysis propels the threading of protein substrates through the central channel, thereby driving unfolding of bound substrates. Substrate threading is mediated by pore loops that are arranged in a spiral staircase and contact backbone residues of a bound substrate in an alternating manner. Structures of multiple AAA+ proteins led to the model that they hydrolyze ATP in a sequential, anti-clockwise manner, thereby propelling the movements of pore loops in multiple steps from the top to the bottom position of the AAA + ring[6,7], threading the substrate. The actual threading mechanism, however, is controversially discussed as substrate threading and pore loop movements appear much faster than ATP hydrolysis and thus remains to be fully elucidated[8,9].

AAA+ proteins functioning in protein degradation thread substrates into an associated or covalently linked peptidase. The interaction between AAA+ and peptidase partners is very stable, allowing for highly processive substrate degradation[1]. The binding of AAA+ proteins to their cognate peptidases involves loop structures (P-loops) or C-terminal tails, which extend from the peptidase-interacting AAA+ ring and dock into hydrophobic grooves (H-sites) located on the surface of the peptidase barrel[10–13]. In case of fused AAA+ and peptidase modules, the connection is mediated by extended linkers[14]. All these flexible elements likely allow for dynamic conformational cycling of the AAA+ ring activity states, while staying bound to the peptidase[14–16].

[1]Science for Life Laboratory, Department of Biochemistry and Biophysics, Stockholm University, Stockholm, Sweden. [2]Center for Molecular Biology of the University of Heidelberg (ZMBH) and German Cancer Research Center (DKFZ), DKFZ-ZMBH Alliance, Heidelberg, Germany. [3]These authors contributed equally: Stavros Azinas, Karin Wallden, Panagiotis Katikaridis. ✉e-mail: a.mogk@zmbh.uni-heidelberg.de; marta.carroni@scilifelab.se; marta.carroni@dbb.su.se

How AAA+ and peptidase partners cooperate during substrate degradation is a key mechanistic question. This is particularly relevant for AAA+ proteases formed by hexameric AAA+ proteins (e.g., bacterial ClpX, ClpA, ClpC) and heptameric peptidases (e.g., ClpP), creating a symmetry mismatch. In the past few years, cryo-EM structures of AAA+ proteases have been shedding light on the communication between these partners[14–25]. Docking of a AAA+ partner opens the entrance gate of the associated peptidase, thereby creating a continuous channel that allows for the transfer of threaded substrates from the AAA+ protein into the proteolytic chamber[13,16,26]. Binding of the bacterial ClpC, ClpX and ClpA AAA+ partners to the ClpP peptidase also stabilizes an extended, active conformation of ClpP harboring a correctly aligned serine peptidase active site[16,19,27]. Together, these findings reveal an allosteric communication between AAA+ proteins and their cognate peptidases, ultimately leading to peptidase activation.

There is also biochemical evidence that peptidases, in turn, signal to their AAA+ partner. ATPase and unfolding activities of *E. coli* ClpA and *M. tuberculosis* ClpC1 and ClpX are enhanced upon binding of the ClpP and ClpP1ClpP2 peptidases[28,29]. The mechanistic basis of AAA+ partner activation by the associated ClpP is unclear, as the cryo-EM structures of the ClpA/ClpP, ClpS/ClpA/ClpP and ClpC1/ClpP1P2 complex do not provide a structural rationale[20,24,25,30,31]. Notably, ClpP association reduces ATPase activity of *E. coli* ClpX, and the reported consequences on ClpX unfolding activity are contradictory[28,30,32–34]. Whether and how peptidases modulate the activities of AAA+ partners remains therefore an unanswered key question.

Here, we determined the cryo-EM structure of the pathogen *Staphylococcus aureus* MecA/ClpC/ClpP complex, a central AAA+ protease of Gram-positive bacteria, at 2.9 Å overall resolution. The ClpC/ClpP protease plays a crucial role in bacterial virulence and has been identified as a drug target[35–38]. *S. aureus* ClpC on its own is inactive and requires cooperation with adapter proteins (e.g., MecA) to form a functional hexamer[39]. Additionally, adapters target specific substrates for degradation by ClpC/ClpP and strongly stimulate ClpC ATPase activity upon substrate transfer[40,41]. As part of a negative feedback loop, MecA itself becomes degraded once the natural target substrate has been processed[40]. The MecA/ClpC/ClpP complex is composed of several layers: (i) a mobile regulatory subcomplex encompassing MecA and interacting with the N-terminal (NTD) and middle (MD) domains of ClpC, (ii) the two ATPase rings of ClpC formed by AAA-1 and AAA-2 domains and (iii) a well-defined double-heptameric ClpP barrel. Interactions between ClpC and ClpP are made by ClpC P-loops and N-terminal β-hairpins of ClpP, which insert into the ClpC channel and contact stretches located next to conserved motifs crucial for ATP sensing and hydrolysis. ClpC/ClpP interactions are dictated by the activity states of the interacting ClpC AAA-2 domain, suggesting a structural pathway for ATPase orchestration and nucleotide sensing by ClpP N-termini. Indeed, ClpP enhances ATPase and unfolding activity of ClpC in an N-terminal β-hairpin-dependent manner. Together, our findings illuminate an interdependent communication between ATPase and proteolytic modules of AAA+ proteases and qualify the ClpP peptidase as an allosteric regulator of its ClpC AAA+ protein partner.

## Results
### Architecture of the *S. aureus* ClpP-ClpC-MecA complex
To determine the cryo-EM structure of the central ClpP/ClpC/MecA protease from *Staphylococcus aureus*, we assembled the components in vitro in the presence of 2 mM ATPγS and assessed complex formation by negative stain. We used the ClpC wild type (WT) as well as the ATPase-deficient ClpC-E280A/E618A (DWB) mutant, which harbors mutations in the Walker B motifs of both AAA-1 and AAA-2 domains, in order to obtain more stable complexes of the mutant. The sample was also supplemented with the *S. aureus* FtsZ protein, known to be a ClpC substrate[42], aiming at visualizing a natural substrate bound to ClpC. The ClpC WT and the DWB mutant gave the same maps, and all reported results here refer to the ClpC WT. The proteins formed mostly double-capped complexes with two AAA

+ ClpC rings loaded onto the ClpP 14-mer (Supplementary Fig. 1a). When imaged under cryo conditions, the assemblies appeared to be preferentially oriented in ice with more end and tilted views over clear side-views (Supplementary Fig. 1a, b). However, iterative rounds of particle picking and 3D classification made it possible to select a subset of 82.854 particles with even Euler angle distribution (Supplementary Fig. 1c). For computational reasons, the reconstruction was focused on a single cap including ClpP (2 x 7mer) : ClpC (6-mer) : MecA (6mer), same as done in studies of similar assemblies (e.g., ClpXP, ClpAP and PAN-CoreParticle[15,17–21,25]).

The overall map of MecA/ClpC/ClpP was refined in CryoSPARCv4.5 to an average resolution of 2.9 Å, but it is resolved at very different levels of detail as shown by the local resolution that ranges from 2 Å in the ClpP part to 9 Å in the MecA-ClpC interaction region (Fig. 1a). High mobility of the ClpC regions mediating MecA binding and, potentially, partial occupancy of MecA explain the lower resolution of this area. Focused refinement, complemented with 3D variability and flexibility analysis, was performed[43,44] by considering the complex organized in three bodies: ClpP (protease body), the ClpC AAA-1 and AAA-2 rings (ATPase body) and "the crown" body composed of the ClpC N-terminal (NTD) and M-domains (MD) bound to MecA (Supplementary Fig. 1d). For the analysis, the bodies were masked in different ways encompassing more or less of the neighboring domains (Supplementary Fig. 1d). In an attempt to push the resolution of each part as much as possible, pertinent symmetry was assigned to each body and symmetry expansion was used when possible (Fig. 1c–e and Supplementary Fig. 1e). As expected, while the ClpP part has an overall D7 symmetry with a very rigid core which can be refined to 2.7 Å resolution, the ClpC ATPase body deviates from a C6 symmetry that cannot really be applied and reaches an overall resolution of 2.9 Å in C1. Interestingly, the resolution of the MecA-crown body could be improved to 3.4 Å when C6 symmetry was imposed (together with signal subtraction) (Supplementary Fig. 1e). By applying C6 symmetry, densities for 3 to 4 MecA N-termini, visible in the C1 map, are lost, and only the MecA C-terminal region can be interpreted at a resolution of ~6 Å and compared to the existing X-ray structures of *B. subtilis* MecA/ClpC[45] (Fig. 2a).

The binding mode of *S. aureus* MecA to ClpC is essentially the same as in *B. subtilis*, suggesting an overall conserved mechanism of regulation of ClpC by MecA across species. More interesting, the unsymmetrized (C1) MecA/ClpC map, combined with AlphaFold2 and 3 (AF2 and AF3)[46,47] predictions, reveals the location of the MecA N-termini relative to the ClpC hexamer (Fig. 2b, c). The prediction of full-length *S. aureus* MecA displays a high-confidence score for the substrate-binding N-terminal domain (NTD) and the ClpC-binding C-terminal domain (CTD) (Fig. 2b) in line with the existing experimental structures of the NTD (pdb code: 2mk6) and CTD (pdb code: 3pxg[45], of *B. subtilis* MecA). The linker region between them is predicted at a low confidence score as a long bent α-helix (Fig. 2b, c), a known artifact of AF[48] when predicting long linker regions. Once the MecA CTD is docked into our experimental MecA/ClpC crown model, different AF conformations place the MecA NTD at various locations relative to the hexamer (Fig. 2d). Of 20 predictions, done with either AF2 or AF3 using either monomeric MecA, MecA/NTD-ClpC 1:1 complex or dimers of MecA/NTD-ClpC (2:2), 8 predictions have MecA NTD inside the ClpC pore, 7 are incompatible with the ClpC hexamer because of MecA NTD clash with ClpC and 5 have the MecA NTD outside the pore, potentially poised for substrate interaction (Figshare chimerax session MecA-ClpC_AF_models.cxs  https://doi.org/10.1101/2025.06.06.658286).  The confidence score of the predictions is similar for all of them, ranging between 0.3 and 0.45.

Interestingly, some of the MecA/NTD-ClpC 2:2 predictions are in good agreement with the extra density present in the MecA/ClpC crown unsymmetrized map (Fig. 2d) and suggest that up to 3 subunits of MecA could position their NTDs at the entrance of the ClpC channel (Fig. 2c, d, and Figshare chimerax session MecA-ClpC_AF_models.cxs https://doi.org/10.1101/2025.06.06.658286). This would imply that substrate-bound MecA NTDs would deliver the substrate to the center of the ClpC channel for unfolding. Using 3D flexibility and 3D variability analysis, which can give

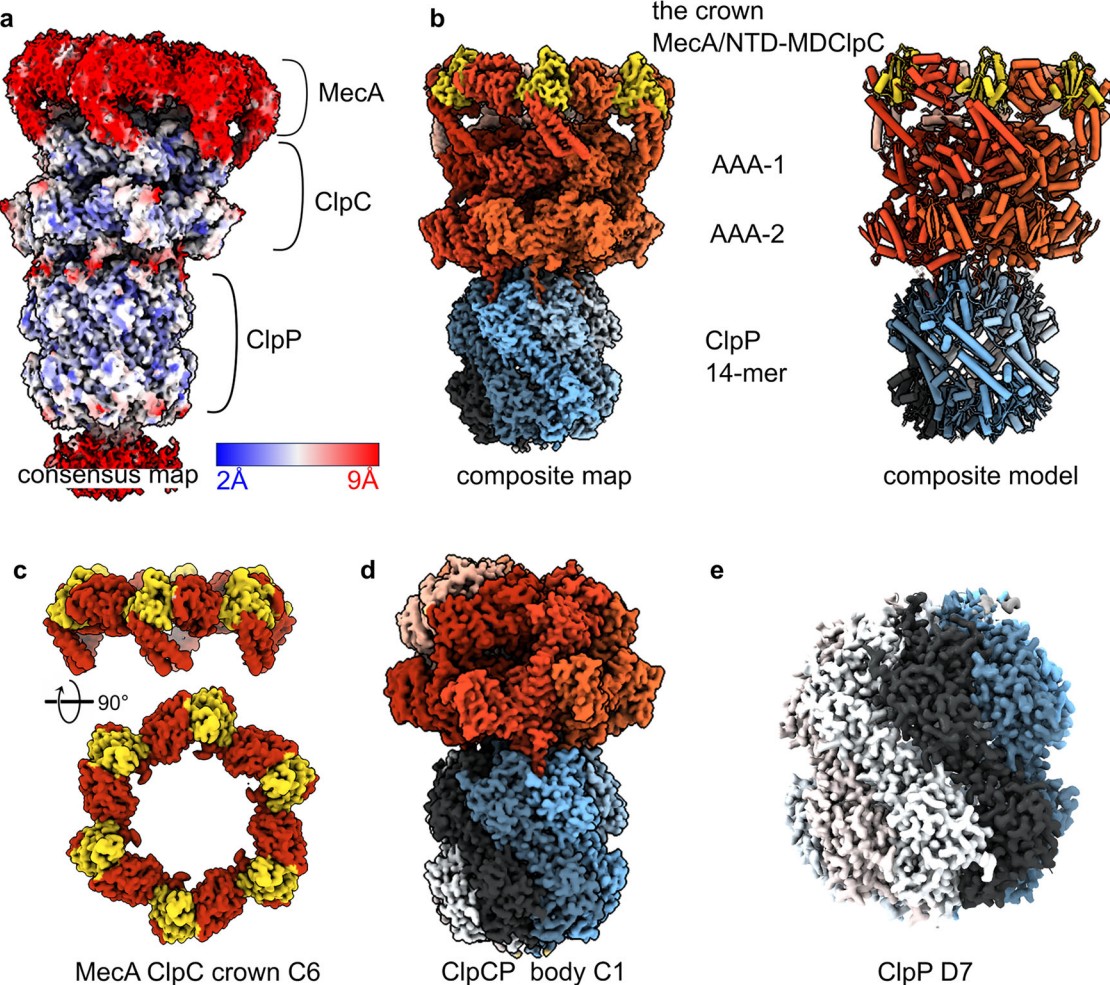

**Fig. 1 | The *S. aureus* MecA/ClpC/ClpP protease complex. a** Consensus cryo-EM map of the complex showing the different protein components and the local resolution. **b** Composite map obtained by combining the map of the MecA crown with the map of the ClpC/ClpP body using Phenix. MecA is colored in gold, ClpC subunits in shades of red and ClpP protomers in shades of blue-gray. **c** Refined map of the MecA crown, which includes the ClpC N-terminal domain (NTDs) and the coil-coiled M-domain (MDs). C6 symmetry was applied. **d** Refined map of the ClpC/ClpP body, which includes the hexamer of ClpC and the tetradecameric ClpP chamber. **e** Refined and D7 symmetrised map of the tetradecameric ClpP chamber.

information about the orientations of a part of the molecule relative to others, it was possible to visualize a screwing movement of the MecA crown relative to the ClpC and ClpP bodies, which suggests a motion for substrate delivery into the ClpC/ClpP chambers (Supplementary Video 1).

After analysis of the MecA crown region, we focused on the analysis of the complex between ClpC and ClpP, and we refrained from applying any symmetry in order to better elucidate the flexible contacts between the AAA + ClpC and the peptidase partner ClpP.

A high-resolution structure of the ClpC/ClpP part of the complex (2.8 Å) was obtained both by masking this body for focused refinement as well as by keeping the MecA crown in the alignment (Supplementary Fig. 1). We refer to this part of the complex as the ClpC/ClpP body. The map was sharpened using different methods (CryoSPARC sharpening, DeepEMhancer[49] and EMReady[50]) without any input model, aiming at gaining detailed information about the interface between ClpP and ClpC.

The *S. aureus* ClpC/ClpP body is made of a compact 7-fold symmetric protease chamber onto which the asymmetric ClpC hexamer sits, tilted similarly to what is observed for other ClpC1P1P2 and ClpA/ClpP complexes[20,24,31]. ClpC forms a shallow right-handed spiral (Fig. 3) with protomer **a** located at the lowest position of the staircase. Two substrate peptides are visible, trapped inside the ClpC channel, each of 8 aa, one grabbed by the AAA-1 pore loops of protomers **a, b, c** and **d** (Fig. 3 and

Supplementary Fig. 2), one by the AAA-2 pore loop of protomers **b, c, d** and **e**. The connection between these two peptides is not resolved, and the quality is not sufficient to derive the amino acid sequence. Accordingly, we only modeled polyA stretches. Subunit **a** is disengaged from the substrate in the AAA-2 ring, and subunit **f** is disengaged from the substrate in both rings. We identify subunit **f**, the most displaced from the ring, as the seam subunit and, together with **a**, it is the least resolved. The *S. aureus* ClpC arrangement is overall similar to the newly reported *S. hawaiiensis* and *M. tuberculosis* ClpC1/ClpP1P2 complexes[24,31] and to the homologous *E. coli* ClpA/ClpP complex[20] or the AAA+ complex ClpX/ClpP[15,16,19]. The number of subunits engaged in contacts with the substrate is different compared to the other complexes (ClpC1P1P2, ClpA/ClpP, ClpX/ClpP), but 4–5 subunits per ring are always substrate-bound.

The contact between ClpC and ClpP is mainly maintained by the ClpC subunits **a-e**, with **b-e** being more tightly packed to one another, better resolved and in closer contact with the ClpP platform (Fig. 4a). Using 3D flexibility and 3D variability analysis, it was possible to visualize a movement of the ClpC spiral relative to the ClpP platform. This movement appears to be a "bowing" of ClpC over the flat ClpP platform (Supplementary Video 1).

The substrate path from the ClpC to the ClpP complex is not straight because ClpC sits offset relative to the proteolytic ClpP chamber entrance (Fig. 3b). If the substrate was threaded rectilinearly down the ClpC channel, it would need to be curved in order to enter the ClpP chamber. Instead, the

**Fig. 2 | The *S. aureus* MecA crown. a** Model of the MecA C-terminal domain (CTD, in gold) sandwiched between two ClpC NTDs and one MD. Comparison between the crystal structure of *B. subtilis* ClpC in complex with MecA (pdb:3xpg) and the one from *S. aureus* solved in this study (pdb:9goq). **b** AlphaFold (AF) prediction of one monomer of *S. aureus* MecA showing a high confidence score for the NTD and the CTD. **c** Comparison of the AF prediction with known experimental structures of MecA NTD (in blue; pdb: 2mk6) and CTD (in gold). **d** AF predictions of MecA alone or with ClpC NTDs with 1:1 or 2:2 stoichiometry (see Figshare chimerax session MecA-ClpC_AF_models.cxs https://doi.org/10.1101/2025.06.06.658286) docked into the cryo-EM C6 MecA crown. Some models are compatible with the hexameric assembly and have the MecA NTD (in blue) placed toward the axial entrance of the ClpC chamber or outside the ring. **e** AF predictions of MecA/ClpC dimers docked into the low-resolution unsymmetrised map of the MecA crown.

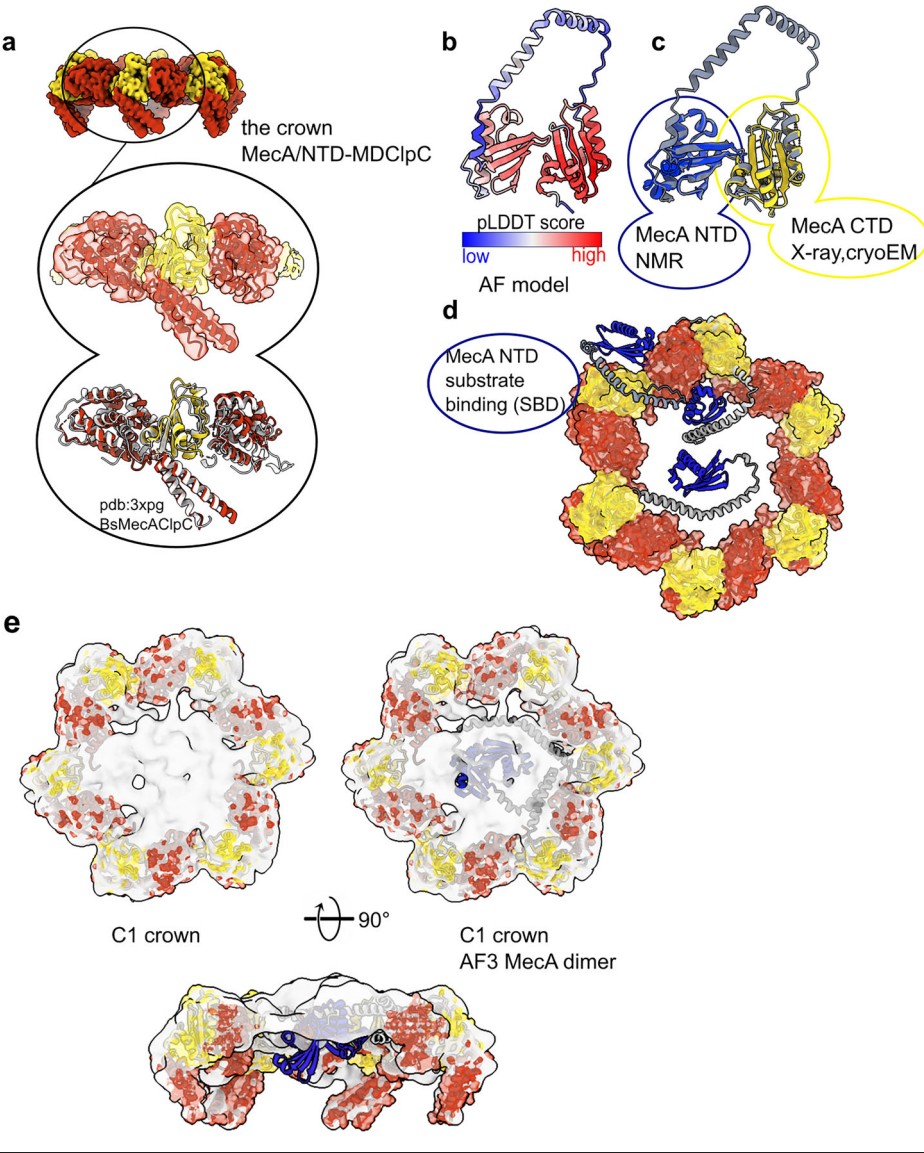

substrate appears to be already threaded obliquely, thus pointing to the entrance of ClpP and possibly directly to one of the catalytic centers (Fig. 3b). This is in accordance with similar observations for ClpX/ClpP[16,19].

The ClpP double heptamer was resolved at 2.7 Å overall resolution (Supplementary Fig. 1e). ClpP is in the extended active conformation with the chamber gate, made of the seven β-hairpins, opened and all catalytic residues (S98, H123 and D172) correctly positioned (Supplementary Fig. 3). The ClpP β-hairpins assume all the "up" conformation extending into the ClpC chamber similar to the crystal structure of *E. coli* ClpP[51] (pdb:1yg6). The pore formed by the β-hairpin collar is ~30 Å in diameter and is much less constrained compared to *E. coli* ClpP crystal structures, being ~15 Å[51].

**The interface between the ClpP barrel and the ClpC spiral**

The interface between ClpP and ClpC is mediated by two main interaction points: the ClpC P-loops and the ClpP N-terminal β-hairpins. The *S. aureus* ClpC P-loops are characterized by an AGF motif that docks into a hydrophobic pocket (H-site) on the outside of the ClpP barrel (Fig. 4b, c). This interaction between bacterial AAA+ unfoldases and the ClpP proteolytic chamber has been extensively described for the ClpX/ClpP and ClpA/ClpP complexes[15,19,20] and more recently for *S. hawaiiensis* and *M. tuberculosis* ClpC1P1P2[24,31], and it has been inferred by a wealth of structural and biochemical studies[10,16,19,20,24,30,51,52].

All six ClpC subunits are in contact with the ClpP platform via the extensive P-loop (residues 660–680) that includes the AGF motif at its tip (residues 671–673). The full extensive P-loops are only visible at a low threshold in the unsharpened map, and in particular, the P-loop of the seam subunit **f** is poorly defined, and the AGF motif is not visible for this subunit, while it is for all the other subunits **a-e** (Fig. 4b, c). This indicates high flexibility of the P-loops as well as differences in the strengths of ClpC P-loop interactions at any given point of the ATPase cycle and the threading activities. This flexibility has been observed for ClpC1/ClpP1P2, ClpX/ClpP and ClpA/ClpP complexes[16,19,20,24,53] and, coupled with the existence of different P-loops and ATPase conformations, has been used to infer a rotational movement of the AAA+ ATPase around to the peptidase chamber. Here, even upon expensive 3D classification, we do not observe alternative conformations of ClpC or of its P-loops, maybe due to the usage of ATPγS rather than ATP or because of the presence of MecA.

The ClpP NTD β-hairpins contact the ClpC inner channel more on one side of the ring involving ClpC subunits **a-e** and ClpP protomers **Pa-Pf** (Fig. 4d). Thus, the asymmetric interaction between ClpP NTD β-hairpins and ClpC partially overlaps with the asymmetry between ClpC P-loops and ClpP H-sites generating a region of tighter contact versus a region of looser one (Fig. 4e). Five out of seven ClpP β-hairpin NTDs (**Pa-Pd, Pf**) are fully visible from residues 1 to 28 while the β-hairpin loop T11-R16 is not visible

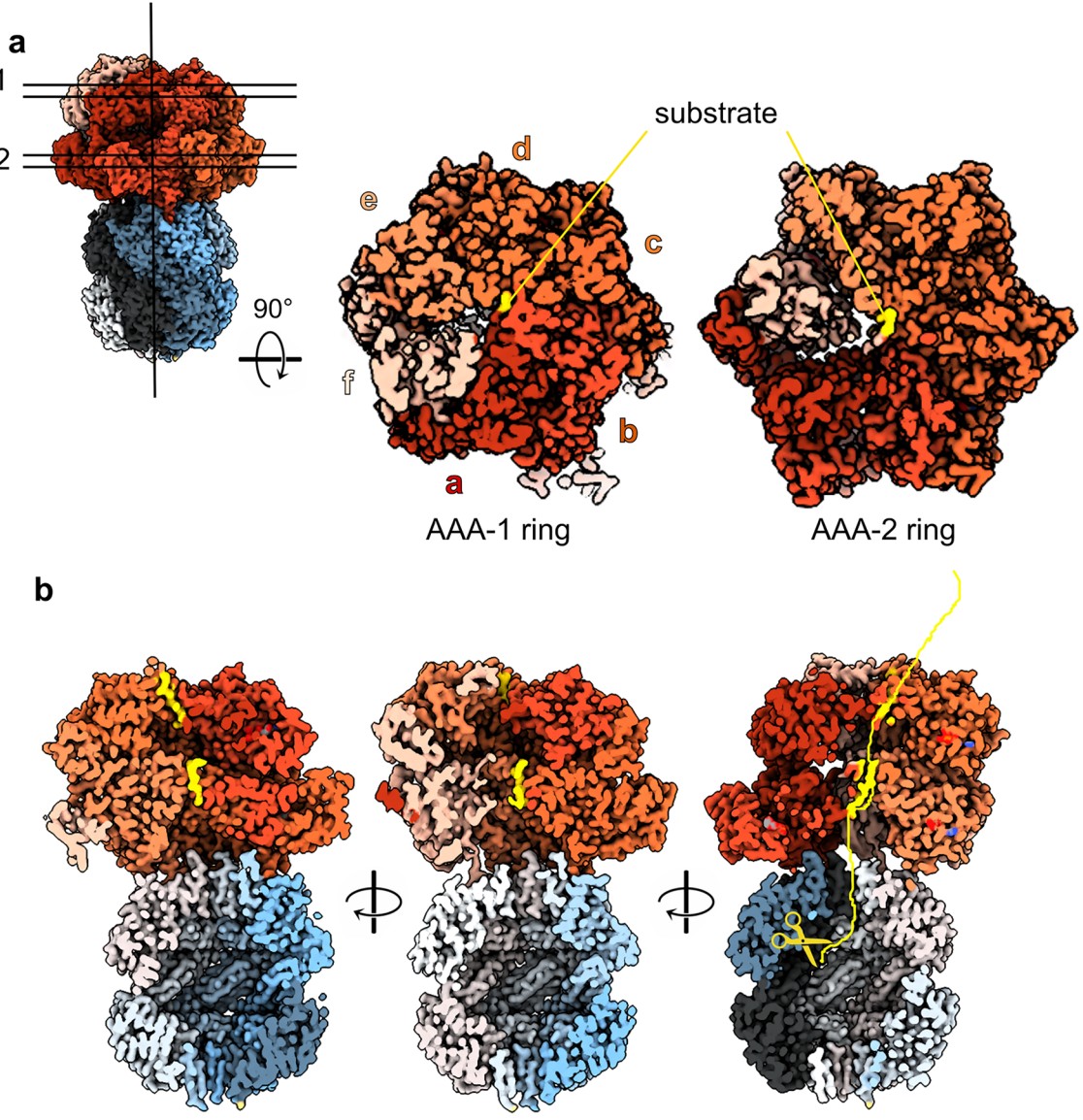

**Fig. 3 | The *S. aureus* ClpC/ClpP body. a** Cut-through of the two ClpC AAA+ rings showing the CpC subunits and their engagement with the substrate (in yellow). **b** Vertical cross-section of the ClpC/ClpP map showing the bound substrate in yellow and its imaginary bent path to one of the proteolytic ClpP centers.

in the ClpP subunits **Pe** and **Pg** (Supplementary Fig. 4). **Pg** in ClpP and **f** in ClpC are not involved in β-hairpin contacts, while they are involved in P-loops contacts, thus contributing to the stability of the ClpC/ClpP complex in the outer parts, but not in the internal channel.

The ClpP β-hairpins contact with the ClpC subunits that are located above them involve different ClpC regions (Fig. 5a, b). One contact involves the ClpC extended P-loop (residues 660–680), and it has also been observed for the single AAA domain of ClpX[16,33] but not for the ClpA AAA-2 domain, which is the closest homologue to ClpC (Supplementary Fig. 5a)[20,25]. We therefore verified the interaction by disulfide cross-linking of ClpP-T11C and ClpC-D667C (Fig. 5c). The efficiency of ClpC-ClpP crosslinking was limited by competing disulfide bond formation between ClpP hairpins. Formation of ClpC-ClpP crosslink products was dependent on MecA, indicating that MecA-driven hexamer formation of ClpC is a prerequisite for ClpC-ClpP interaction (Fig. 5c).

The other two ClpC regions contacted by ClpP β-hairpins are located close to the ATPase centers of the AAA-2 domains. One region comprises two helices, the long α22-helix (residues Y682 to S696; Supplementary Fig. 5a) and the short α23-helix, from P699 to R704, which is the R-finger (Fig. 5b, Supplementary Fig. 5a). We call it the allosteric R-region (allo-R) as

we reason that the contacts involving this area may have an effect in repositioning the R-finger. We confirmed the contact between ClpP β-hairpins and the allo-R by disulfide crosslinking of ClpP-R13C and ClpC-S696C, which were again dependent on MecA presence (Fig. 5d). A second region, made of residues 620–624, is in close proximity to the Walker B motif of the ClpC AAA-2 domain, and we call it the allosteric W-region (allo-W) (Fig. 5a, b; Supplementary Fig. 5a).

Interestingly, the interactions between the ClpP N-terminal β-hairpins and these ClpC regions (P-loops, allo-R and allo-W) vary around the ATPase ring in a way that might suggest a direct regulation of the ClpP subunits into the ClpC ATPase centers. Contacts between ClpP and ClpC can be described as "walking" from the allo-W to the P-loops, along the allo-R helix α22 (Fig. 5e). ClpC subunit **a** exhibits limited contacts (~6 Å) with the β-hairpins of **Pb** and **Pa** mediated by the tip of the long allo-R α22 helix (residue K694) and the allo-W (residue K621). Adjacent ClpC subunit **b** displays the closest interactions (3–4 Å) between the tip of allo-R α22 helix (residues S696 and P699) and the underneath **Pd** and **Pc** β-hairpins. Moreover, subunit **b** also contacts tightly β-hairpin **Pc** residue N12 via the allo-W residue E620. ClpC subunits **c** and **d** interact with **Pd-Pe-Pf** β-hairpins via the central part of the allo-R α22 helix (residues N695, K691,

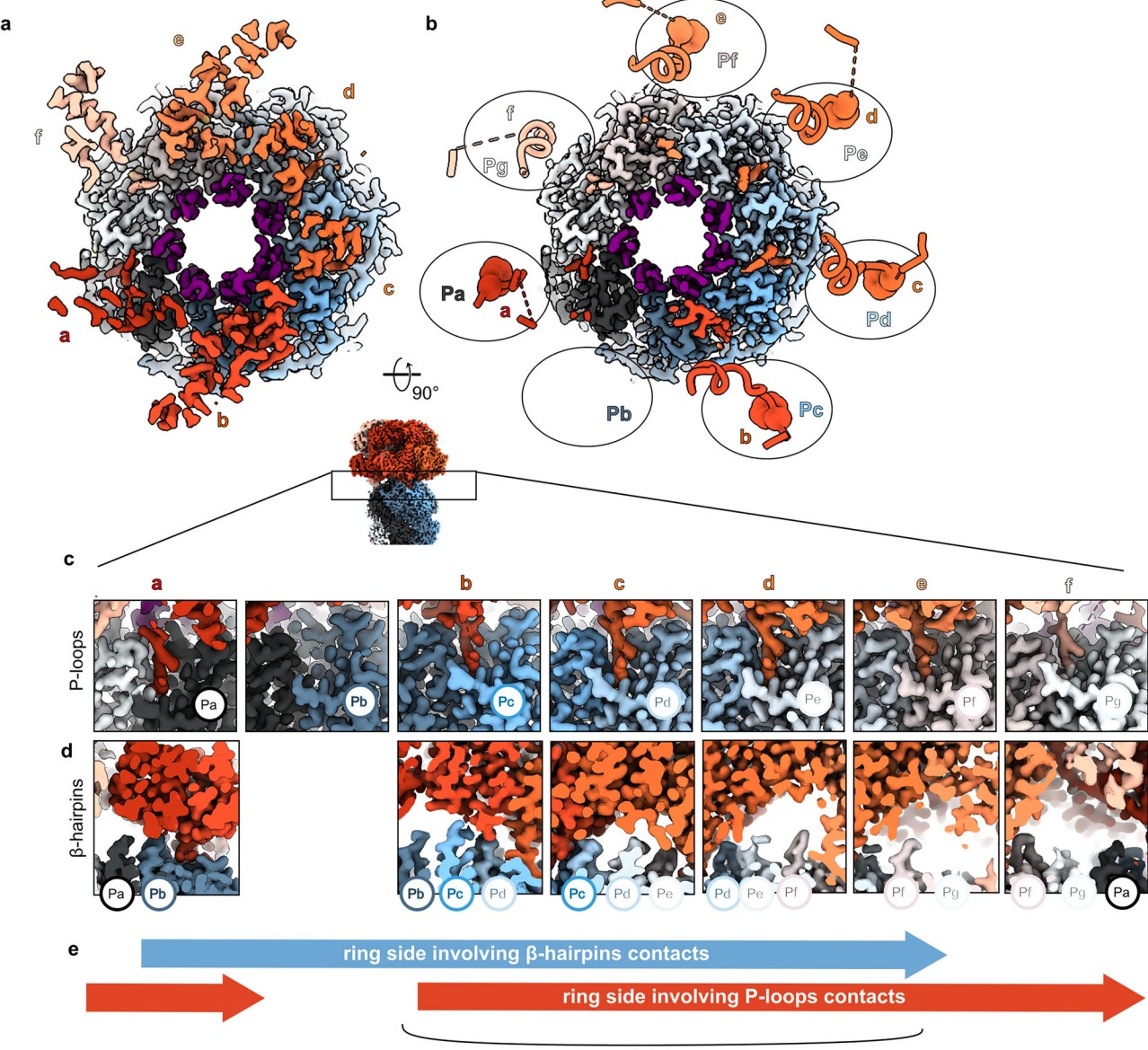

**Fig. 4 | The ClpC/ClpP interface. a** Cryo-EM map of the ClpC/ClpP interface with the ClpP β-hairpins (residues 1–20) colored in dark purple. **b** ClpC/ClpP interface with the cryo-EM map of ClpP and the ribbon and atom model representation of ClpC extended P-loops (residues 620–680) with the AGF motifs in spheres and not visible for subunit f. The different ClpC subunits and ClpP protomers are displayed with the same color code as in Fig. 1 and throughout the manuscript. **c** Cut through the cryo-EM map to visualize the details of the interactions between each ClpC subunit's P-loop and ClpP protomers' H-sites. **d** Cut through the cryo-EM map from the interior of the complex chamber to visualize the details of the interactions between each ClpC subunit's AAA-2 domain and the ClpP protomers' β-hairpins. **e** Scheme showing the contacts around the ring between ClpC and ClpP.

E692), and they additionally display interactions with P-loops residues Q663 and E664. To follow, ClpC subunit **e** only makes contacts via the distal part of the allo-R α22 helix (residue T684) and P-loop residues. Protomer **f** does not interact at all with the ClpP β-hairpins underneath (**Pa** and **Pf**) (Fig. 5e).

On the ClpP side, the β-hairpin residues T10, T11, N12 and R13 mediate the contact with the ClpC subunits sitting above (Fig. 5e). Briefly, the pattern of interactions between the ClpP β-hairpins and the ClpC subunits walks along the allo-R α22 helix to be shifted from the ATP center and the allo-W to the allo-R α22 distal part and the P-loop, thus describing a possible molecular path of communication from the ATP center to the P-loops that anchor the ClpC hexamer onto the ClpP proteolytic chamber.

The ClpP hairpins also form stabilizing contacts between each other via T6, R13, G14, E15, R16 and additionally S22, a residue that is not part of the β-hairpin (Supplementary Fig. 4b). These lateral ClpP β-hairpin contacts involve the loop of the hairpin for protomers **Pa-Pf**, while they shift toward the base of the β strand for the **Pf-Pg-Pa** interaction. Lateral contacts between neighboring ClpP β-hairpins have also been observed in the ClpA/ClpP and the *M. tuberculosis* ClpX/ClpP complex, though involving different residues[20,54].

## ClpP β-hairpins play crucial roles in oligomerization and ClpC interaction

We tested for the functional relevance of ClpP β-hairpins in ClpC interaction and cooperation during substrate degradation by mutating the β-

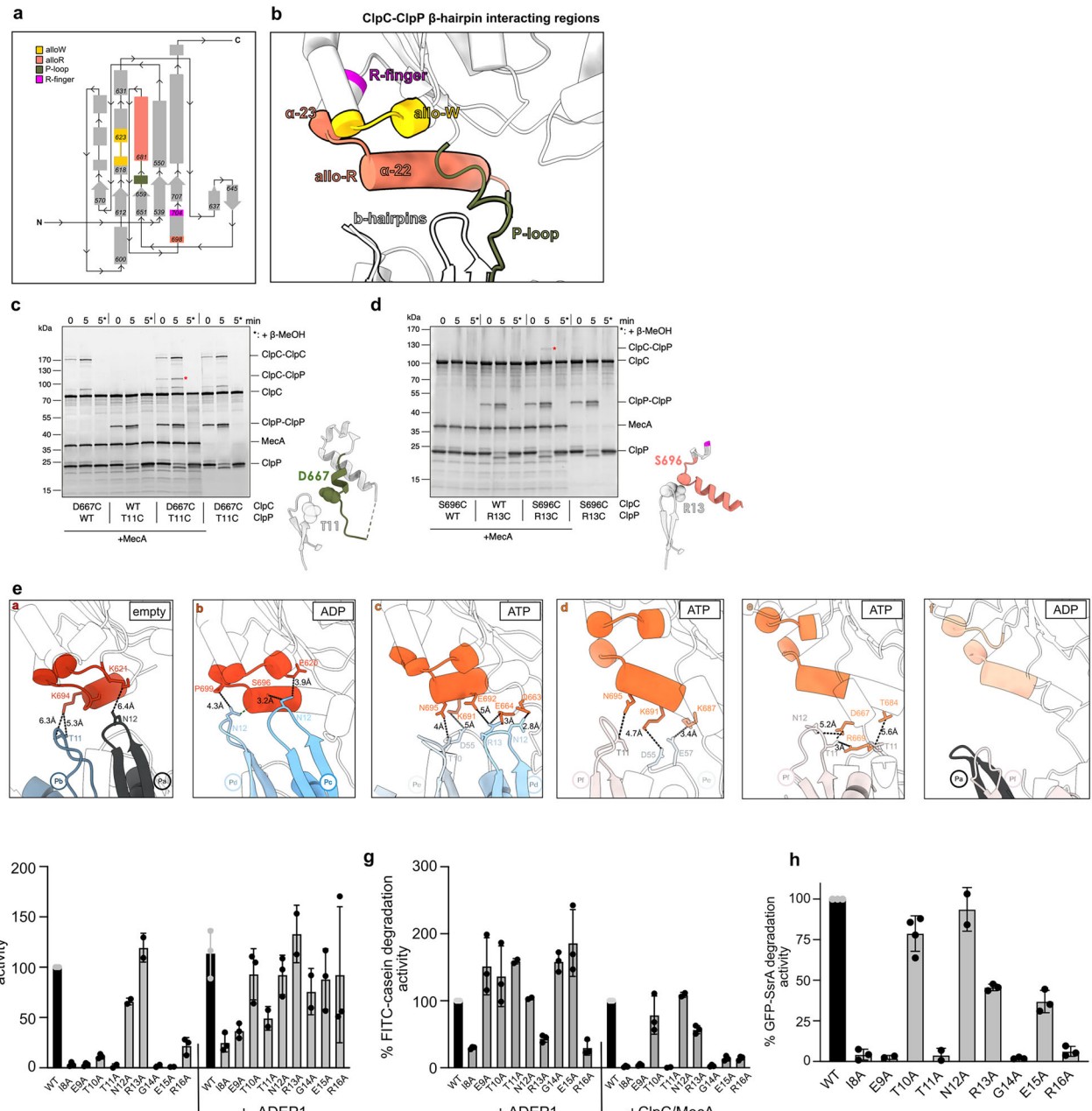

**Fig. 5 | Contacts between ClpP β-hairpins and ClpC AAA-2 domains affect degradation activity. a** Topological scheme of the ClpC AAA-2 domain involved in contacts with the ClpP β-hairpins. To facilitate the visualization, each region involved in contacts is indicated and colored. **b** Ribbon model of the same regions labeled and colored according to (**a**). **c**, **d** Disulfide crosslinking experiments of neighboring residues of the ClpP β-hairpin and ClpC residues of the extensive P-loop (**c**) and allo-R region (**d**). ClpC wild type (WT), ClpC-D667C, ClpP wild type (WT), ClpC-T11C and MecA were incubated in the presence of 2 mM ATPγS as indicated. Disulfide crosslinking was induced under oxidizing conditions (+Cu(Phe3)). Presence of β-mercaptoethanol (β-MeOH) in the SDS sample buffer is indicated (*). Crosslink products were analyzed by SDS-PAGE. Positions of ClpP-ClpP and ClpC-ClpP disulfide crosslinks are indicated. **e** Subunit by subunit,

visualization of the contacts between ClpP β-hairpins residues and ClpC residues of the allo-W, allo-R and P-loops regions. The sequence of contacts from ClpC subunit **a** to subunit **f** around the ClpC AAA-2 ring shows a shift of contacts from one tip to the other of the long allo-R helix. **f** LY-AMC degradation was monitored in the presence of ClpP wild type (WT) and indicated mutants in the absence or presence of ADEP1. Degradation rates were determined and set to 100% for ClpP WT (-ADEP1). **g** FITC-casein degradation rates by ClpP/ADEP1 or MecA/ClpC/ClpP (WT or mutants) were determined. Degradation activities of ClpP WT reactions were set to 100%. **h** Degradation rates of GFP-SsrA by MecA/ClpC/ClpP (WT or mutants) were determined. Degradation activity of ClpP WT was set to 100%. Error bars represent standard deviations ($n = 3$) (**f**–**h**).

hairpins residues from I8 to R16 to alanines. We first determined proteolytic activities of ClpP mutants toward the peptide substrate LY-AMC, a degradation reaction that does not require ClpC assistance. Only ClpP-N12A and R13A showed wild-type-like degradation activity toward LY-AMC, whereas all other ClpP mutants were either entirely or largely inactive (Fig. 5f, Supplementary Table 1). Addition of the ClpP activating drug ADEP1 largely restored peptide degradation by ClpP β-hairpin mutants,

though the activities of some mutants (e.g., I8A, E9A) remained reduced (Fig. 5f). ADEP1 mimics AAA+ partner docking by binding to ClpP H-sites and induces long-range conformational changes, including opening of the ClpP entry pores and interactions between ClpP monomers and heptameric rings[38,55]. This stabilizing effect can explain why ADEP1 binding compensates for structural defects of ClpP β-hairpin mutants, which was also observed for the *S. aureus* ClpP V7A mutant[56]. ADEP1 also allowed most

mutants to degrade the fluorescently labeled unfolded model substrate FITC-casein except for I8A, R13A and R16A, which showed reduced activity compared to ClpP WT (Fig. 5g, Supplementary Table 1).

The *S. aureus* ClpP-V7A mutant forms an inactive split-ring conformation in the absence of ADEP[56]. In the same study, the β-hairpin mutations P5G, V7A, and I20V showed evidence of unfolding, as well as general ClpP ring instability. We therefore probed for structural integrity of ClpP β-hairpins mutants and monitored formation of tetradecameric barrel-like complexes by EM and size exclusion chromatography (SEC) (Supplementary Fig. 6). Only ClpP-T10A, N12A and R13A formed barrel-like tetradecameric structures in EM and showed a similar elution profile as ClpP-WT in SEC runs. ClpP-E15A and R16A also showed the formation of rings/barrel-like structures in EM, but to a much lesser extent than ClpP-WT (Supplementary Table 1).

To probe for defects of ClpP β-hairpin mutants in complex formation with ClpC, we again employed negative stain EM and SEC (Supplementary Fig. 7). EM visual analysis showed that ClpP-N12A, R13A and E15A formed proteasome-like particles with ClpC in the presence of MecA and ATPγS similar to ClpP WT (Supplementary Fig. 7a). Other β-hairpin mutants showed reduced (E9A, T10A and T11A) or severely affected (I8A, G14A, R16A) complex formation with ClpC/MecA and oligomeric particles observed in EM largely represent MecA/ClpC complexes. Similar findings were made when studying MecA/ClpC/ClpP complex formation by SEC. Here, the interaction between ClpC and ClpP was transient and resulted in a shift of ClpP to earlier elution volumes but no co-elution with ClpC (Supplementary Fig. 7b). This is explained by the presence of ATP during the SEC run, allowing for ATP hydrolysis and MecA degradation, ultimately causing dissociation of the ClpC/ClpP complex. A shift of ClpC elution volumes was not observed when using the ClpC P-loop mutant G672D/F673D, which does not interact with ClpP, documenting specificity (Supplementary Fig. 7b). Elution profiles of ClpP-T10A, N12A and E15A in the presence of ClpC/MecA were similar to ClpP WT, while ClpP-I8A, E9A and G14A hardly showed ClpC complex formation, consistent with EM analysis. ClpP-T11A, R13A and R16A elution volumes were shifted in the presence of MecA/ClpC-WT, though less pronounced as compared to ClpP WT (Supplementary Table 1).

We next analyzed the abilities of ClpP β-hairpin to cooperate with MecA/ClpC in the degradation of unfolded FITC-casein and GFP-SsrA. All ClpP hairpin mutants except N12A showed defects in MecA/ClpC-dependent protein degradation (Fig. 5g, h). For ClpP-I8A, E9A, T11A, G14A and R16A these activity defects correlate with their severe structural defects and deficiency in complex formation with ClpC/MecA (Supplementary Table 1). ClpP-T10A and R13A and E15A mutants exhibited partial activities, and the severity of defects was dependent on substrate identity. Degradation of GFP-SsrA was approximately 2.5-fold reduced for R13A and E15A mutants, while proteolysis of FITC-casein was much more affected for ClpP E15A (14% activity compared to ClpP WT). Notably, FITC-casein degradation could be restored for most ClpP mutants in the presence of ADEP1, but not MecA/ClpC (Fig. 5g), indicating a specific role of the β-hairpins for ClpC-mediated activation of ClpP.

Taken together, these findings indicate a crucial role of ClpP β-hairpins for ClpP oligomerization, MecA/ClpC/ClpP complex formation and proteolytic activities with strongly differing effects of individual point mutations (Supplementary Table 1). Our data define three groups of mutants, which (i) are similar to ClpP WT (N12A), (ii) show strong defects in complex formation (I8A, E9A, T11A, G14A, R16A) and (iii) are proficient in ClpC binding, though showing reduced degradation activities in ClpC-dependent assays (T10A, R13A, E15A). This hints at a function of ClpP β-hairpins in the communication with the ClpC partner.

### Coordination of ClpC-ClpP contacts and activity states of interacting AAA-2 domains

The specific defects of some ClpC β-hairpins mutants in ClpC-dependent proteolysis and the analysis of the contact between the ClpP β-hairpins and ClpC protomers around the AAA-2 ring point to a role as allosteric elements modulating ClpC ATPase activity. Type of contacts and their strengths differ for the individual β-hairpins (Fig. 5e), and we therefore analyzed whether these differences in ClpC-ClpP interactions are linked to distinct activity states of ClpC AAA-2 domains. The resolution of the ClpC/ClpP body structure is adequate to precisely determine the nucleotide state of most ATP pockets apart from the AAA-2 of protomer **a** (Fig. 6a).

We found that the asymmetry of the ClpP/ClpC interface is also reflected in activity states of the ClpC AAA-2 ring (Fig. 6a, b). The ClpC subunit **a**, located at the bottom of the AAA-2 spiral, is likely empty as only residual density for the adenosine ring is visible at low threshold and the R-finger from **f** is far away (>10 Å). However, it has to be considered that the local resolution for subunits **a** and **f** is lower. Subunit **b** is ADP-bound, and even if some small extra density seems to be present, which could be a $Mg^{2+}$ or a leaving Pi, it has the donating **a** R-finger far away (~9 Å), and there is not enough density to properly fit ATPγS; it is therefore defined as inactive. Subunits **c-e**, have ATPγS bound with visible $Mg^{2+}$, R-fingers within active distance (between 2 and 3 Å) and all WalkerB residues canonically placed; **c-e** are therefore all defined as active. The seam **f** is in the ADP-bound state, thus inactive. In summary, the AAA-2 ring can be separated into an inactive half, including subunits **f-b** and an active half represented by ATPase-poised subunits **c-e** (Fig. 6b). The nucleotide states of the AAA-2 subunits suggest that ATP hydrolysis propels in a counter-clockwise manner around the ClpC ring, in agreement with findings for other AAA+ proteins[20]. Subunits **c** and **d**, which are expected to hydrolyze ATP next, are also the ones that bind the substrate most tightly (Supplementary Fig. 2). The distribution of AAA-2 activity states suggests that ATP hydrolysis and substrate threading are coupled, as previously observed.

In the ClpC AAA-1 ring, the nucleotide states are similar to AAA-2, yet they are shifted relative to AAA-2 by one subunit (**a**: ADP, **b-e** ATP bound, **f**: empty; Fig. 6a). This suggests that the AAA-1 and AAA-2 rings work in alternating ATPase cycles. This had also been observed for the disaggregase ClpB and the ClpA/ClpP complex[20,57] to guarantee a tight grip on the substrate[20,57] and avoid backsliding.

As shown above, the interactions between ClpP β-hairpins and ClpC AAA-2 domains differ in their contact sites and strengths (Figs. 4, 5, Supplementary Fig. 5) and have a shifted pattern compared to the ATPase activity states. ClpP β-hairpins establish a tight contact with the subunits at the interface between the active and the inactive half of the AAA-2 ring. In particular, ClpC subunit **b** is contacted by the ClpP **Pd** and **Pc** β-hairpins in contacts (<4 Å) that involve simultaneously the allo-W and allo-R regions. This suggests the existence of a possible mechanism of sensing an inactive subunit (**b**) and repositioning the R-finger for an adjacent active one (**c**). Looking at the overlay between all AAA-2 ATP pockets, we observed the repositioning of residue N659, which is highly conserved across ClpC, ClpA and ClpE (Supplementary Fig. 5a), in ATPγS-bound ClpC protomer **c**. This residue is interesting because it is located at the beginning of the P-loop, and it is facing the three WalkerB residues (allo-W residue E620 and Walker B residues D617 and E618) (Fig. 6c). By looking at the pattern of interactions involving the ClpC protomers **b** and **c** at the active/inactive interface of the AAA-2 ring, we can identify possible paths of communication between ClpP and the ClpC AAA-2 active site. The ClpP β-hairpin (protomer **Pc**) that contacts the allo-W region and the tip of the allo-R α22-helix can in one hand position the **b** R-finger so that ATP hydrolysis can happen in **c** and at the same time pull on the **c** P-loop, potentially re-arranging the ATP pocket while keeping it anchored to ClpP (Fig. 6d). This pattern only exists between **b** and **c** and not between **e-f**, thus giving processivity to the hydrolysis events.

### ClpP enhances ATPase and unfolding activities of ClpC via its N-terminal β-hairpins

To probe for an impact of ClpP on ClpC function, we determined ClpC ATPase activity in the absence and presence of saturating ClpP levels. We focused on those ClpP mutants that do not exhibit oligomerization defects and form a complex with ClpC. ClpP-WT increased the ATPase activity of MecA/ClpC complexes by 1.4-fold from 45 to 65 ATP/min/monomer (Fig. 7a). This stimulatory effect was reduced for R13A and E15A, which still

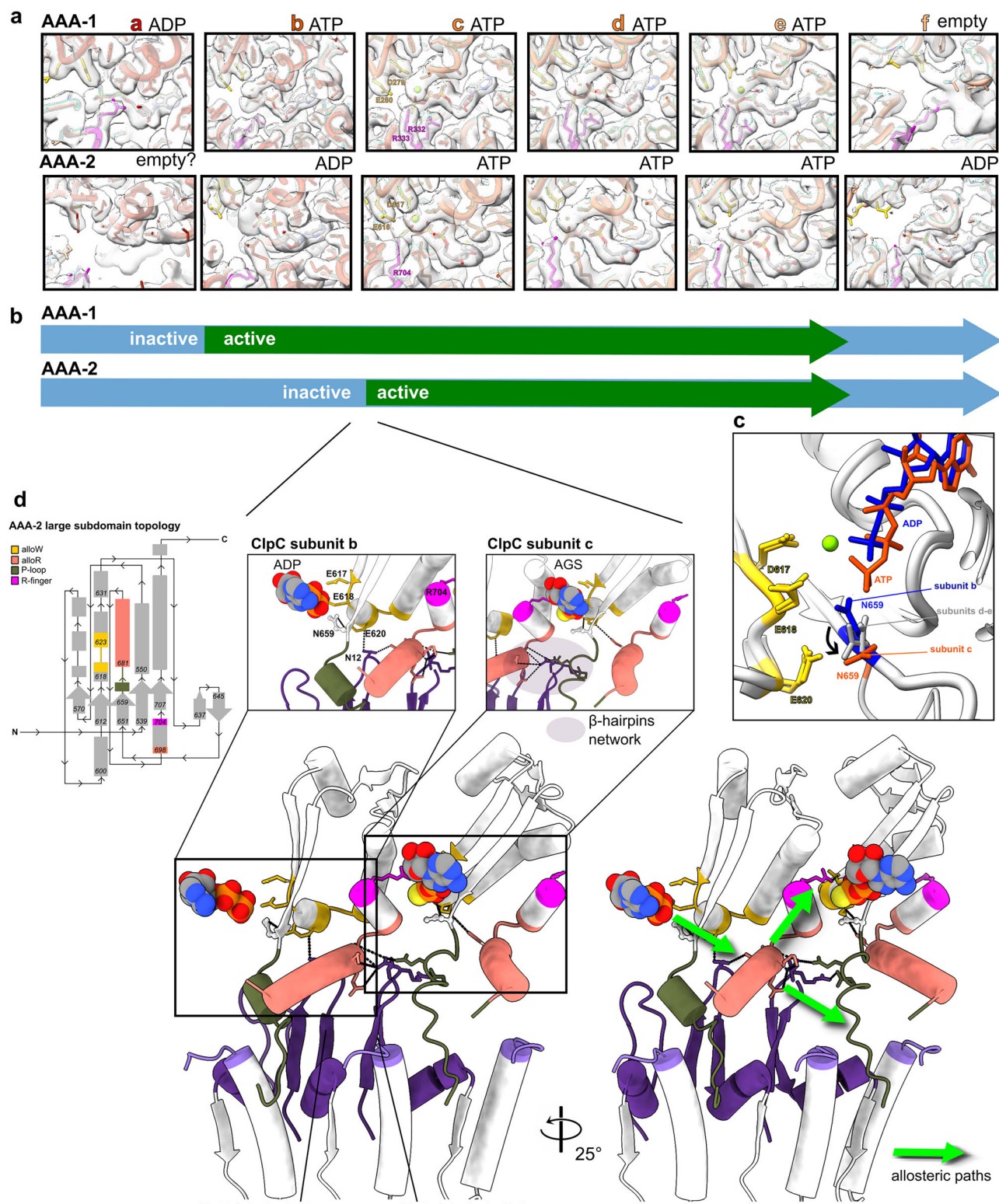

**Fig. 6 | ClpP-ClpC contacts and AAA-2 ATPase activity states. a** Nucleotide-binding state of the ClpC subunits. The subunits are shown as ribbon liquorice. The nucleotides are shown as sticks colored by element; the Mg atom is in lime green, the R-fingers and the WalkerB residues are shown as sticks in magenta and yellow, respectively. The modeled nucleotide is ATPγS, but indicated as ATP for simplicity. Cryo-EM density is shown. **b** Schematic of the ATPase active (green) and inactive (light blue) subunits around the AAA-1 and AAA-2 rings. **c** Superimposition of subunits **b** (ADP-bound, dark blue) and **c** (ATP-bound orange) on the AAA-2 ATP pocket showing the position of Walker B residues D617 and E618, as well as the position of allo-W residue E620 and conserved N659. N659 reorients itself from the

ADP to the ATP-bound state. N659 is positioned at intermediate positions in subunits **d**, **e**, and it is not fully visible for subunit a–f. **d** Interface between the inactive subunit **b** and active subunit **c** in the AAA-2 ATP pocket. Topological scheme in the top left corner is shown to facilitate the localization of the various regions. The **Pc** β-hairpin contacts of allo-W of subunit **b** and the distal part of the allo-R helix subunit **b** that is also contacted by the **Pd** β-hairpin, which in turn also contacts the P-loop of subunit **c**. This forms a β-hairpin network, which could define two allosteric paths toward the ATP pocket of subunit c and the P-loop/H-site region of ClpC to ClpP anchoring.

**Article**

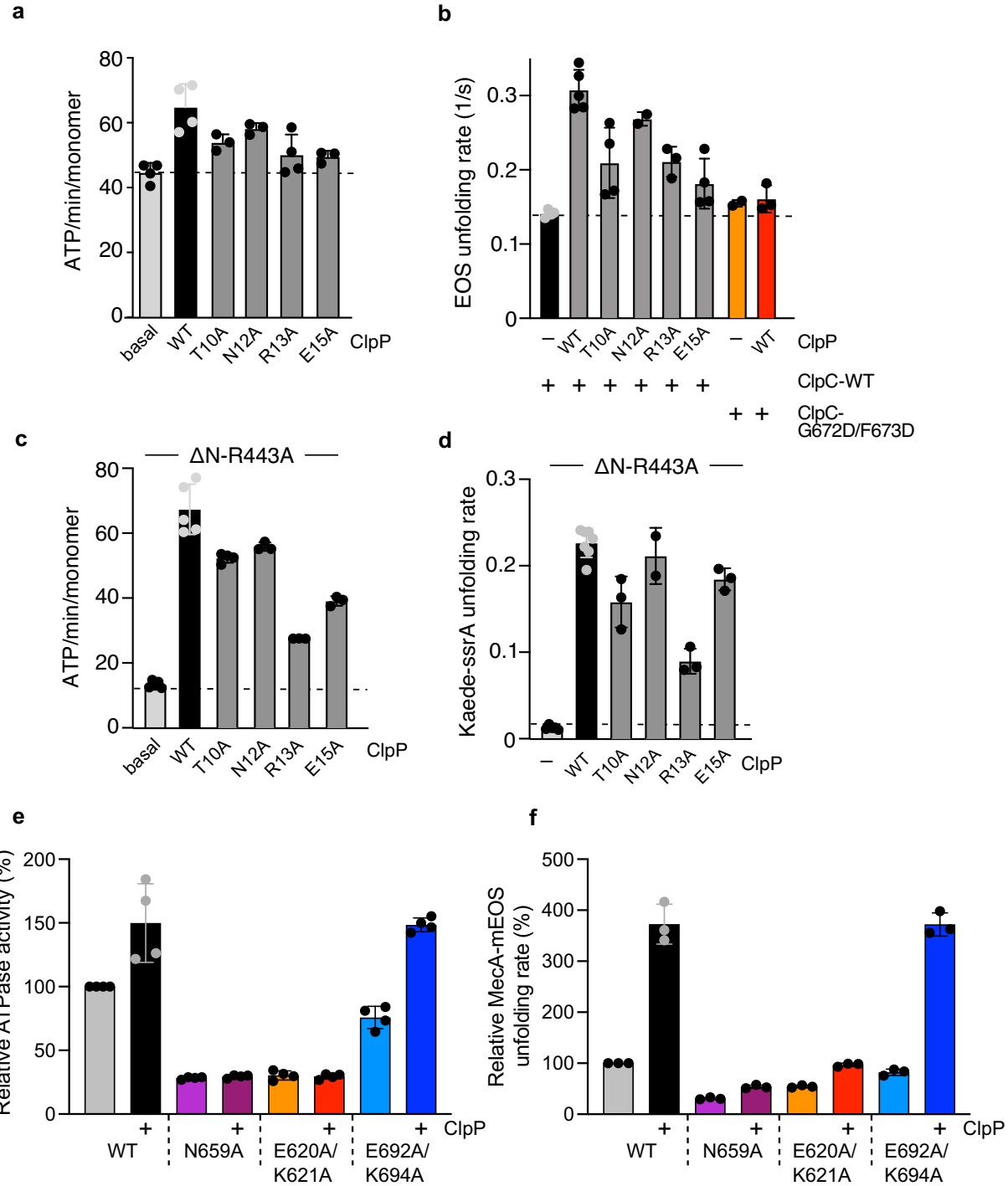

**Fig. 7 | ClpP β-hairpin mutants are affected in stimulating ClpC ATPase and unfolding activities. a** ATPase activities of MecA/ClpC complexes were determined in the absence and presence of ClpP wild type (WT) or β-hairpin mutants. **b** Unfolding of MecA-mEOS3.2 by ClpC was monitored in the absence or presence of ClpP WT and indicated β-hairpin mutants. Initial MecA-mEOS3.2 fluorescence was set as 100% and MecA-mEOS3.2 unfolding rates were determined. **c** ATPase activities of ΔN-ClpC-R443A were determined in the presence of ClpP WT or mutants as indicated. **d** Unfolding of Kaede-SsrA by ΔN-ClpC-R443A was monitored in the absence or presence of ClpP WT and β-hairpin mutants. Initial Kaede-

SsrA fluorescence was set as 100% and unfolding rates of ΔN-ClpC-R443A were determined. **e** ATPase activities of ClpC WT (+MecA), N659A and alloW (E620A/K621A) and alloR (E692A/K694A) mutants were determined in the absence and presence of ClpP. The activity of ClpC WT (+MecA) was set to 100%. **f** Unfolding rates of MecA-mEOS3.2 by ClpC WT, N659A and alloW (E620A/K621A) and alloR (E692A/K694A) mutants were determined in the absence and presence of ClpP. Unfolding activity of ClpC WT was set at 100%. Error bars represent standard deviations ($n = 3$) (**a**–**f**).

interact with ClpC (Fig. 7a, Supplementary Table 1). We speculated that the increase in ATPase activity also enhances ClpC threading and unfolding activities. This could also explain reduced proteolytic activities of ClpP-R13A and E15A, as they stimulate ClpC ATPase activity to a lesser extent as compared to ClpP-WT. To directly monitor ClpC unfolding activity, we made use of the fluorescent fusion constructs MecA-mEOS3.2 and Kaede-SsrA, which are substrates of ClpC. MecA-mEOS3.2 includes only the C-terminal domain of MecA (A91-E239), which is sufficient for ClpC binding, ATPase activation and targeting for autodegradation[58,59]. The fluorophors mEOS3.2 and Kaede were photoactivated, resulting in single cleavage events of the polypeptide chains and preventing the fluorophors from refolding after initial unfolding by ClpC[60–62].

MecA-mEOS3.2 is rapidly unfolded by ClpC, indicating that the fusion construct acts as a bona fide adapter (Fig. 7b). Loss of MecA-mEOS3.2 fluorescence was approximately 2-fold faster in the presence of ClpP-WT as compared to unfolding reactions containing ClpC only, demonstrating that ClpP enhances ClpC threading activity (Fig. 7b, Supplementary Table 1). The threading activity of the ClpC P-loop mutant G672D/F673D was not stimulated by ClpP-WT, documenting specificity. Addition of ClpP-T10A, R13A and E15A resulted in stimulated degradation of MecA-mEOS3.2 (Supplementary Fig. 8a), however, to minor degrees, in accordance with reduced ClpC ATPase activation by these mutants (Fig. 7b). These findings suggest that ClpP β-hairpins contribute to the enhanced ClpC ATPase and unfolding activities upon ClpC/ClpP complex formation.

When monitoring Kaede-SsrA unfolding by MecA/ClpC, we only observed a partial loss of fluorescence in the presence of ClpP but no decrease without the peptidase (Supplementary Fig. 8b). This is likely caused by MecA acting as competing substrate, explaining (i) incomplete Kaede-SsrA degradation in the presence of ClpP due to MecA autodegradation (Supplementary Fig. 8c) and (ii) protection of Kaede-SsrA from unfolding in the absence of ClpP. To circumvent ClpC dependence on MecA, we made use of a ClpC mutant that does not require MecA for activation and autonomously degrades stable model substrates: ΔN-ClpC-R443A. The ClpC M-domain residue R443 plays a crucial role in ClpC repression by providing a key contact for resting state formation[39]. Mutating this residue causes constitutive ClpC-R443A hexamer formation, enabling MecA-independent proteolytic activity toward unfolded FITC-casein. The additional deletion of the N-terminal (N) domain enables ΔN-ClpC-R443A to even degrade stable GFP-SsrA. We determined that the ATPase activity of ΔN-ClpC-R443A is 5-fold stimulated by ClpP and is almost identical to the one determined for MecA/ClpC/ClpP complexes (Fig. 7c; Supplementary Fig. 9a). This mechanistically explains why ΔN-ClpC-R443A is proficient in degrading stable substrates and enabled us to study the impact of ClpP on ΔN-ClpC-R443A unfolding activity using Kaede-SsrA as substrate. ClpP presence strongly enhanced ΔN-ClpC-R443A unfolding activity by 20-fold (Fig. 7d, Supplementary Fig. 9b). ClpP-T10A, N12A and particularly R13A and E15A partially increased ΔN-ClpC-R443A ATPase activity (Fig. 7c). ClpP-T10A, R13A and E15A also stimulated unfolding activity, though to a lower degree as compared to ClpP WT (2-fold, 3-fold and 1.4-fold reduction for T10A, R13A and E15A, respectively) (Fig. 7d). These ClpP mutants allowed for complete degradation of Kaede-SsrA (Supplementary Fig. 9c), indicating that the slower kinetics of Kaede-SsrA fluorescence loss stem from reduced activation of ΔN-ClpC-R443A unfolding capacity.

Together, these findings indicate that ClpP activates ATPase and unfolding activities of ClpC with contributions of its N-terminal β-hairpin.

In a reciprocal approach, we probed for the relevance of ClpC allo-W and allo-R regions for the stimulatory effects of ClpP by analyzing E620A/K621A (allo-W) and E692A/K694A (allo-R) mutants. Additionally, we mutated N659, which our structural analysis implicates in signaling ClpP association to the ATPase center (Fig. 6c). The ClpC N659A and the allo-W but not the allo-R mutant exhibited reduced degradation activity toward FITC-casein (Supplementary Fig. 9d). This can be rationalized by a reduced ATPase activity of ClpC-E620A/K621A and ClpC-N659A that is additionally not stimulated upon ClpP binding (Fig. 7e). Accordingly, ClpP presence only modestly increased the reduced unfolding activity of the

mutants toward MecA-mEOS3.2 (Fig. 7f, Supplementary Fig. 9e). In contrast, ATPase and unfolding activities of the allo-R mutant E692A/K694A were strongly increased by ClpP (Fig. 7e, f, Supplementary Fig. 9e). These findings indicate that the integrity of the ClpC allo-W site and N659 are crucial for ATPase and unfolding activation by the ClpP partner.

## Discussion
### Overall organization and dynamics of the *S. aureus* MecA/ClpC/ClpP AAA+ protease
**Adapter protein MecA NTDs wards the entrance of the ClpC unfoldase chamber.** Our structure of the *S. aureus* MecA/ClpC/ClpP AAA+ protease is one of the first of an AAA+ unfoldase in complex with its peptidase (ClpP) and an adapter protein (MecA). A structure of the ClpS/ClpA/ClpP has been presented[25] with the adapter ClpS visible when the map is filtered at low resolution. In the MecA/ClpC/ClpP structure as well, MecA and the interacting ClpC N-terminal (NTD) and middle domains (MD) are resolved only at low resolution (6–9 Å), unless C6 symmetry is imposed (Fig. 2a). We refer to this apical part of the structure, including MecA and the interacting ClpC NTD and MD as the "MecA crown" region and we observe, via 3D flexibility analysis that the crown is very dynamic (Supplementary Video 1) and it undergoes a screwing motion suggesting a way of accompanying the substrate inside the ClpC channel. Interestingly, when 6-fold symmetry is applied, details up to 3.4 Å can be obtained, which are in perfect agreement with the crystal structure of the *B. subtilis* ClpC AAA-1 domain in complex with the C-terminal domain of MecA[45] (Fig. 2b). By applying symmetry, density for the MecA NTD is however lost. The non-symmetrised map, displayed at low threshold, shows instead density for MecA NTDs, capping the full complex (Fig. 2d). This density, combined with Alpha-Fold predictions of multimers of ClpC and full-length MecA, suggests that up to 3 copies of MecA could have simultaneously their NTDs positioned at the top entrance of the ClpC channel. This positioning enables MecA to control substrate access to ClpC and, consistently, its NTD governs substrate specificity[59]. The regions involved in adapter protein binding and substrate recruitment of the unfoldases, such as ClpC or ClpA, are extremely interesting. However, detailed, near-atomic information that would give clear insight into their regulatory mechanism is still missing, and each adapter-AAA+ protease complex likely works in a very specific and unique manner.

### *S. aureus* ClpC hexameric arrangement and substrate binding are similar to homologous AAA+ unfoldases
The arrangement of the *S. aureus* AAA+ protease MecA/ClpC/ClpP structure presented here shows an overall arrangement similar to the one previously observed for *M. tuberculosis* and *S. hawaiiensis* ClpC1/ClpP1P2 complexes[24,31,54], *E. coli* ClpA/ClpP[20,25] and *E. coli*, *M. tuberculosis* and *N. meningitidis* ClpX/ClpP[15,16,54]. The complex is characterized by asymmetry, where a fairly static and 7-fold symmetric ClpP barrel functions as a platform for an asymmetric ClpC hexamer. The spiral arrangement of the ClpC subunits is similar to the one observed in cryo-EM structures of *B. subtilis* and *M. tuberculosis* ClpC alone[23,63], in line also with other AAA+ unfoldases such as ClpB and Hsp104[64–66], the Rpt1-6 part of the 26S proteasome[17], the archael PAN complex[21], as well as the already mentioned ClpX and ClpA hexamers[15,16,20,25]. ClpC in the MecA/ClpC/ClpP complex binds a peptide substrate in its central channel via pore loops of both the AAA rings (Fig. 3). In general, the ClpC subunits that are engaged with the substrate are tightly packed to one another and better resolved as in similar AAA+ protease assemblies, indicating that substrate binding stabilizes the complex. In the AAA-1 ring, subunits **a-d** bind to the substrate peptide and are in their ADP, ATP, ATP, ATP state, respectively. In the AAA-2 ring **b-e** pore loos bind to the substrate and are in ADP, ATP, ATP and ATP state, respectively (Fig. 3, Supplementary Fig. 2 and Fig. 6a). The exact number and subunits involved in substrate interactions and their ATP state vary in the different AAA+/ClpP complexes (ClpA/ClpP, ClpX/ClpP and ClpC1/ClpP1P2), thus suggesting that there is not a very

precise and determined coupling for each and every ATPase subunit. This does not fully agree with the hand-over-hand mechanism considered until recently the process for substrate threading. For instance, in *S. hawaiiensis*, ClpC1/ClpP1P2 subunits are all in the ATP state regardless of being substrate-engaged or not[24]. Structural observation of substrate bound even to all-ADP subunits has been recently reported also for the Lon protease[67]. In the *M. tuberculosis* ClpC1/ClpP1P2 structure, ATP binding to the top subunit is suggested to drive substrate interaction[54], however, in our structure the top subunit **e** of the AAA-1 ring is active (and ATP bound) but does not contact substrate (Supplementary Fig. 1). It is therefore difficult to draw specific conclusions on the threading mechanism based on cryo-EM snapshots. Our structure shows a defined order of inactive and active AAA-1/AAA-2 subunits, which suggests that a high degree of coordination within and between AAA-1 and AAA-2 rings is crucial for high ATPase and threading activities.

**Flexible interactions between ClpC and ClpP allow for ClpC subunit movements while maintaining complex integrity.** The main contact between ClpC and ClpP is mediated by long and flexible ClpC P-loops, consistent with findings for AAA+Clp/ClpP complexes[15,16,19,20,24,54]. The flexibility of the P-loop will allow ClpC subunits movements during its ATPase cycle while keeping hold on ClpP. P-loops contain a specific ΦGF (AGF in *S. aureus* ClpC) motif that docks into H-sites on the external surface of the ClpP barrel[10,33,68] (Fig. 4b). Accordingly, upon mutation of the AGF motif (G672D, F673D; Supplementary Fig. 7b), ClpC/ClpP interaction is lost. In the MecA/ClpC/ClpP complex, all six ClpC P-loops are in contact with ClpP, with one H-site left empty (Fig. 4b, c). This pocket is placed between ClpC subunit **a** (empty ATP pocket) and ADP-bound subunit **b** (Figs. 4b, 6a). The AGF motifs of the ClpC P-loops, or part of it, are visible for five ClpC subunits (Fig. 4b), but not for the seam subunit **f**. As expected, this indicates a reduced interaction strength between subunit **f** and ClpP (Fig. 4e). This increased flexibility likely enables large movements of this seam subunit **f**. Still, the remaining part of the extended P-loop (residues 660–680) works as a large and flexible interaction surface maintaining contact to ClpP. The various levels of stretching for the P-loops of the different ClpC protomers illustrate their role in accommodating ClpC subunit movements and, accordingly, have been suggested to work as shock absorbers[53]. Cryo-EM analysis of the P-loops positionings in the PAN/CP[21], ClpX/ClpP[20] and ClpA/ClpP[20], ClpC1/ClpP1P2[24,54] complexes, generating different conformations of the AAA+ unfoldase and of the P-loops, was used to infer a rotational movement of the AAA+ hexamer on the ClpP platform. Crosslinking of ClpA P-loops with the interacting ClpP pockets, however, did not abolish degradation activity, indicating that rotation of the AAA+ unfoldase on the peptidase platform is not required for substrate degradation, although a certain level of ClpA mobility is likely necessary[69]. Here, based on 3D flexibility analysis, we observe that ClpC bows relative to the ClpP barrel platform (Supplementary Video 1). A rotational movement of ClpC over ClpP is not obvious from this analysis, and even upon extensive 3D classification, focused on the ClpC/ClpP interactions area, we were not able to obtain different P-loop binding conformations.

**ClpC and ClpP function as reciprocal allosteric activators**
Allosteric communication between ATPase unfoldases and their partner ClpP peptidase has been described for several AAA + /ClpP protease systems. In particular, the role played by the AAA + P-loops in the activation of ClpP has been extensively described for ClpX[30,70,71]. As observed in the structures of all the other AAA+ /ClpP complexes[19,27] also *S. aureus* ClpC binding stabilizes ClpP in an active extended conformation with a correctly aligned peptidase catalytic triad (Supplementary Fig. 3). Compared to the ClpP only structure, ClpC also induces a widening of the proteolytic entrance gate to ~28 Å, similar to ClpX/ClpP, ClpA/ClpP[16,20,53] and ADEP-bound ClpP[38], creating an open translocation channel. The occupancy of the ClpP H-sites thus serves as an allosteric signal to trigger peptidase activation[72] in the *S. aureus* MecA/ClpC/ClpP AAA+ protease.

A reciprocal regulation from the ClpP peptidase to the AAA+ unfoldase has also been observed, but with in parts opposite results and unclear molecular mechanisms. In the *E. coli* ClpX/ClpP, ClpP has been reported to inhibit ClpX wild-type ATPase activity while enhancing ATPase activity of specific ClpX mutants[30]. *M. tuberculosis* ClpP1P2, stabilized by the activator benzoyl-leucine-leucine (Bz-LL), increases the ATPase activity of both ClpX and ClpC1[71], and, similarly, ATPase activity of *E. coli* ClpA is enhanced upon ClpP interaction[28,29,73]. In *S. aureus*, we also observe that ClpP enhances ClpC ATPase and unfolding activities. This effect is observed for MecA-activated ClpC and, even stronger, for an activated ClpC mutant (ΔN-R443A), which functions independently of the adapter (Fig. 7a–c; Supplementary Fig. 9). Moreover, our analysis of ClpP NTD β-hairpin loop mutants shows that they display different degrees of defective cooperativity with ClpC. Defects in AAA+ partner cooperation have also been reported for β-hairpin mutants of human mitochondrial ClpP[74] and *E. coli* ClpP[51]. Some of the *S. aureus* ClpP mutants exhibit structural defects, indicating a crucial role of the β-hairpins for ClpP barrel integrity (Supplementary Fig. 7). Interestingly, mutation of R16, which is involved in the majority of ClpP β-hairpin contacts between adjacent ClpP protomers (Supplementary Fig. 4b), displays serious defects in complex assembly, pointing at an important stabilizing role of β-hairpins in the formation of functional *S. aureus* ClpP barrels. Other ClpP mutants (T10A, R13A, E15A) are affected in ClpC activation, while still forming a complex with ClpC (Supplementary Fig. 7, Supplementary Table 1).

Our structure provides a possible structural rationale for these observations and for the allosteric activation of ClpC by ClpP. ClpP β-hairpins interactions with ClpC AAA-2 vary around the ring and involve two sites of the ClpC AAA-2 domain located next to the ATP pockets, the allosteric-W (allo-W) region and allosteric-R (allo-R) region, and the extended P-loops (Figs. 5a, b, 8). The allo-W region is directly coupled to the AAA-2 ATP pocket with residue E620 being aligned with the Walker-B residues E618 and D617 (Fig. 6c). The allo-R region is composed of a helix-turn-helix motif with one long helix α22, which gets contacted by the ClpP β-hairpins at different positions around the ring (Figs. 5c, 8) like a sliding rod, and the second short helix α23 that contains the R-finger. Our mutational analysis highlights the relevance of the allo-W region for an increase in ClpC ATPase and threading activities upon ClpP binding (Figs. 5f–h, 7a–d). The tightest contacts are observed between ClpP β-hairpins of protomers **Pc** and **Pd** and AAA-2 of subunits **b** and **c** (Fig. 8), thus at the interface between a subunit that has just undergone ATP hydrolysis and one that could hydrolyze ATP next. This points to a function of β-hairpins in nucleotide sensing, as also postulated in the past for *E. coli* ClpP[33]. The network of ClpP β-hairpins interactions at the **b-c** interface suggests a possible pathway of molecular interactions that would sense the ADP state of one subunit and reposition the R-finger and the P-loops to facilitate ATP hydrolysis on the adjacent ATP-bound subunits (Fig. 6d). This would support sequential directionality of the ATPase around the ring and determine which ClpC subunit (in this case, **c**) should undergo hydrolysis. Interestingly, ClpP reduces cooperativity between ClpA AAA domains during substrate threading[73], which supports the idea that ClpP β-hairpins promote a more sequential mode of ATP hydrolysis by the AAA+ partner. The biochemical characterization of the ClpP β-hairpin mutants supports, in part, the predicted role, as ClpP-T10A and R13A association results in reduced stimulation of ClpC activity. The defects of ClpP-E15A can be explained by E15 making lateral contacts to adjacent β-hairpins, and its mutation might alter correct β-hairpin positioning. Surprisingly, ClpP-N12A exhibits only minor defects, despite N12 contacting allo-W and allo-R regions of ClpC. Our structure only provides a snapshot of the functional ClpC/ClpP cycle, and the interactions between the ClpP β-hairpins and ClpC subunits might partially vary depending on the actual step of the cycle, explaining the weak phenotype of these mutants.

In summary, our structural and biochemical analysis of the MecA/ClpC/ClpP complex from *S. aureus* highlights the interdependent communication between the ATPase and the peptidase components of this AAA + protease. The network of interactions between ClpP and ClpC that varies

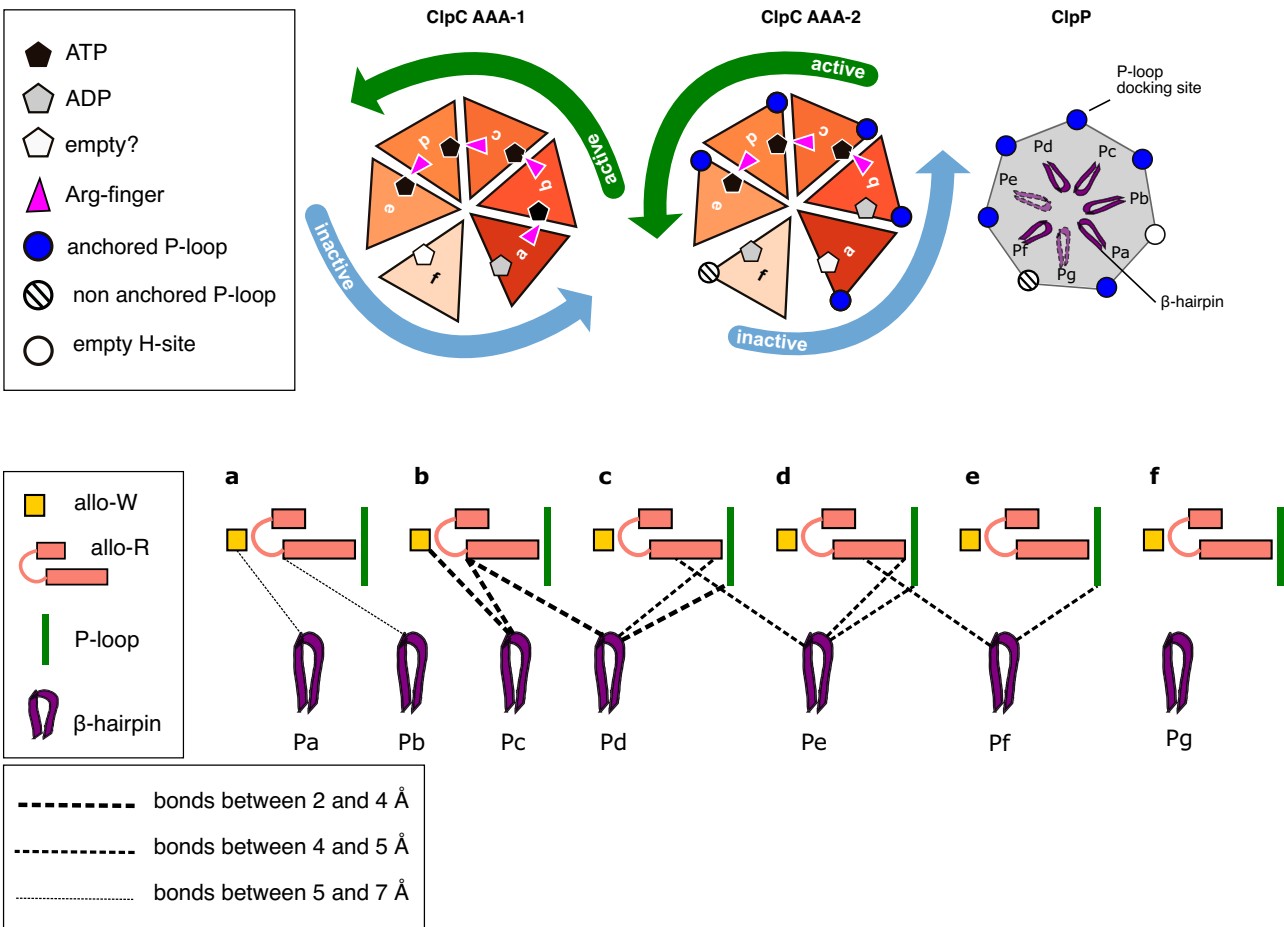

**Fig. 8 | Summarizing schematic showing (upper panels) the ClpC ATPase states in each ring, the state of the ClpP H-sites receiving the ClpC P-loops and the visibility of the ClpP β-hairpins (light versus dark purple).** In the bottom panel, for each ClpC subunit (a–f), the general contacts between the AAA-2 ring ClpC allosteric regions and ClpP β-haipins are represented.

around the ring can provide a rationale for the allosteric effect of ClpP over ClpC.

## Methods

### Strains, plasmids and proteins

*E. coli* strains (Supplementary information) used were derivatives of MC4100, XL1-blue or BL21. GFP-SsrA, MecA-mEOS3.2, ClpC, MecA and ClpP were expressed from pDS56, yielding N- or C-terminal His$_6$-tag fusions. Kaede-SsrA was expressed as fusion construct harboring N-terminal His$_6$-SUMO. ClpC and ClpP mutant derivatives were generated by PCR mutagenesis. PCR products were digested with DpnI, transformed into XL1-blue cells and mutations were confirmed by sequencing of isolated plasmids.

ClpC (wild type and variants), MecA and ClpP (wild type and variants) were purified after overproduction from *E. coli* ΔclpB::kan cells. His$_6$-GFP-SsrA was purified after overproduction from *E. coli* ΔclpX ΔclpP cells. His$_6$-SUMO-Kaede was overproduced in *E. coli* BL21 cells. All proteins (wild type or mutants) were purified using Ni-IDA (Macherey-Nagel) following the instructions of the manufacturer. ClpC, MecA and ClpP were next subjected to size exclusion chromatography (Superdex S200, GE Healthcare) in 50 mM Tris pH 7.5, 25 mM KCl, 10 mM MgCl$_2$, 5% (v/v) glycerol, 2 mM DTT. The His$_6$-SUMO tag of Kaede-SsrA was cleaved off by His$_6$-Ulp1 addition in 50 mM Na-phopshate pH 8.0, 300 mM NaCl, 5 mM β-mercaptoethanol and Kaede-SsrA was isolated by passing the mixture through a Ni-IDA column.

MecA-mEOS3.2 and Kaede-SsrA were photoactivated by exposure to UV light (395 nm, Morpilot flashlight) or sunlight for 2 h, respectively.

Efficiencies of photoactivations were calculated to 45% for both fluorophors by determining absorbance at the following wavelengths: MecA-mEOS3.2: ε507: 63,400 M$^{-1}$ cm$^{-1}$, ε572: 32,200 M$^{-1}$ cm$^{-1}$; Kaede-SsrA: ε508: 98,800 M$^{-1}$ cm$^{-1}$, ε572: 60,400 M$^{-1}$ cm$^{-1}$.

Pyruvate kinase of rabbit muscle and FITC-casein were purchased from Sigma. Protein concentrations were determined with the Bio-Rad Bradford assay.

### Cryo-EM specimen preparation and data acquisition

Protein samples were prepared as described in ref. 40. Complex of *S. aureus* ClpC(WT)-MecA-ClpP-FtsZ-ATPgS was formed by incubating a mixture of 4 μM ClpC-DWB, 5 μM MecA and 4 μM ClpP in reaction buffer (100 mM HEPES pH 7.4, 10 mM KCl, 10 mM MgCl$_2$, 2 mM DTT and 2 mM ATPγS) for 5–10 min at 35 degrees, and thereafter put on ice. A volume of 3 μl mixture was loaded onto Quantifoil R2/2 holey carbon grids (Cu 300 mesh, Electron Microscopy Sciences) that were glow-discharged for 60 s at 20 mA using a GloQube (Quorum) instrument. The grid was flash-frozen using a Vitrobot (Thermo Fischer Scientific FEI Mark IV). The samples were visualized at liquid nitrogen temperature on a Titan Krios G2 electron microscope (Thermo Fischer Scientific FEI) operating at 300 kV equipped with a Gatan BioQuantum energy filter and a K3 Summit direct electron detector (AMETEK). Cryo-EM images were recorded at a nominal magnification of ×105,000 corresponding to a calibrated pixel size of 0.828 Å. The total exposure time was adjusted as such so that it resulted in an accumulated dose of ~40 electrons/Å$^2$ and a total of 40 frames in each image stack. In total, 12,990 images were collected using EPU automatic data collection with defocus values ranging from 0.8 to 2.4 μm.

## Cryo-EM image processing, 3D reconstructions and model building

The cryoSPARC v4.1+ software package was used with the packaged patch motion correction, contrast transfer function (CTF) estimation, blob picking and two-dimensional classification performed for an initial selection of random 250 micrographs. Then, several rounds of supervised particle picking, via Topaz training[75], using 3D classification as an indicator of improvement in picking particles, were performed.

While MecA/ClpC/ClpP oligomers mostly had preferred end view and tilted view orientations, the multiple rounds of picking refinement allowed us to include side and other views (Supplementary Fig. 1). Due to computational limitations, the extracted particles had a corresponding pixel size of 1.06 Å. This produced a consensus structure with Non-Uniform[76] refinement with global CTF correction, at a reported 2.88 Å resolution.

We then performed multiple rounds of masked signal subtraction and focused refinements of the different regions of MecA/ClpC/ClpP oligomers. To produce the 3D masks, we used ChimeraX to mask around the regions of interest. For the refinement around the MecA crown region, we applied a C6 symmetry, which improved the local resolution dramatically compared to the consensus density map. Similarly, for the ClpP region, we applied a D7 symmetry, which improved the local resolution to 2.68 Å (Supplementary Fig. 1d). Reported resolutions are based on the gold standard Fourier shell correlation (FSC) using the criterion of 0.143. All the obtained maps were postprocessed using EMReady[50]. A composite map using the best-aligned parts was created using Phenix[77].

Model building was performed starting from monomers generated with AlphaFold2[47], followed by chain refinement in Coot[78] and then further model building and refinement alternating cycles of Coot, Phenix[77] and Isolde[79]. The nucleotide pockets were initially built using Alpha_Fill[80] and then further refined in Phenix and Isolde[77,79]. Model building was aided by the EMReady solvent-flattened maps, but all final model refinement was performed against unsharpened maps in Phenix. Maps and models' statistics are reported in Table 1.

## Biochemical assays

### ATPase activity.
Rates of ATP hydrolysis were determined by a coupled-colorimetric assay linking ADP production to the formation of lactate from pyruvate. Reduction of pyruvate is linked to the oxidation of NADH to NAD$^+$, which was monitored by measuring the absorbance at a wavelength of 340 nm. The assay was carried out in buffer A (50 mM Tris pH 7.5, 25 mM KCl, 10 mM MgCl$_2$, 2 mM DTT) containing 2 mM ATP and an ATP regenerating system (PK/LDH mix: 250 μM NADH, 500 μM PEP, 1/20 (v/v) PK/LDH (Sigma-Aldrich)) at 30 °C using a CLARIOstar$^{plus}$ plate reader (BMG Labtech). The final protein concentrations were as follows: ClpC (0.6 μM), MecA (0.9 μM) and ClpP (up to 1.2 μM). ATPase rates were determined for at least three biological replicates based on the linear decrease of NADH absorbance at 340 nm and were calculated according to the following equation:

$$\text{ATPase rate} = \frac{1}{\varepsilon_{\text{NADH}} \times C_{\text{ClpC}} \times d} \times \frac{\Delta A_{340nm}}{\Delta t}$$

where $\varepsilon_{\text{NADH}}$ is the molar absorption coefficient of NADH at a wavelength of 340 nm (6220 M$^{-1}$cm$^{-1}$); $C_{\text{ClpC}}$ is the final concentration of ClpC (0.6 μM); d is the optical path length (1 cm); $\Delta A_{340\,nm}/\Delta t$ is slope of the linear decline in absorption at a wavelength of 340 nm.

### Degradation assays.
200 μM LY-AMC (Succinyl-L-Leucyl-L-tyrosyl-AMC) was incubated with 1 μM ClpP in buffer A in the absence or presence of 1.5 μg/ml ADEP-1. LY-AMC proteolysis was monitored by determining LY-AMC fluorescence intensity using 355 nm and 460 nm as excitation and emission wavelengths, respectively.

FITC-casein, GFP-SsrA, MecA-mEOS3.2 and Kaede-SsrA degradation assays were analyzed using a LS55 spectrofluorimeter (Perkin Elmer) in buffer A at 30 °C. The final protein concentrations were as follows 0.3 μM FITC-casein, 0.5 μM GFP-SsrA, 0.5 μM MecA-mEOS3.2, 0.5 μM ClpC, 0.75 μM MecA, 1 μM ClpP. In case of Kaede-SsrA (0.5 μM) degradation/unfolding assays, 1.5 μM ClpC (wild type or ΔN-ClpC-R443A), 2.25 μM MecA and 3 μM ClpP were used. All assays were carried out in the presence of an ATP regenerating system (0.02 mg/ml PK, 3 mM PEP pH 7.5) and 2 mM ATP. ADEP1 was used in LY-AMC and GFP-SsrA degradation assays at a concentration of 10 μg/ml. The increase of FITC-casein fluorescence upon its degradation was monitored by using 490 and 520 nm as excitation and emission wavelengths, respectively. Loss of GFP-SsrA, MecA-mEOS3.2, Kaede-SsrA fluorescence was recorded using 400 and 510 nm (GFP-SsrA), 560 nm and 580 nm (MecA-m-eOS3.2) and 540 nm and 582 nm (Kaede-SsrA) as excitation and emission wavelengths. MecA-mEOS3.2 and Kaede-SsrA degradation was additionally analyzed by SDS-PAGE followed by staining with SYPRO® Ruby Protein Gel Stain (ThermoFisher).

LY-AMC and FITC-casein degradation rates were calculated based on the initial slopes of fluorescence signal increase. GFP-SsrA, MecA-mEOS3.2, Kaede-SsrA degradation/unfolding rates were determined by fitting the data to single exponential decay using Prism 6.0 (GraphPad). Initial fluorescence intensities were set to 100%. Rates were determined based on at least three independent experiments, and standard deviations were calculated.

### Size exclusion chromatography.
Complex formation of ClpC (6 μM), MecA (9 μM) and ClpP (12 μM) was monitored by size exclusion chromatography (SEC, Superose 6 10/300 GL, GE Healthcare) in buffer A containing 5% (v/v) glycerol and 2 mM ATP at room temperature. Proteins were first incubated for 5 min in the presence of 2 mM ATPγS prior to injection. Fractions were collected in 96-well plates, aliquots taken and subjected to SDS-PAGE. Gels were stained using SYPRO® Ruby Protein Gel Stain (ThermoFisher).

### Disulfide crosslinking.
Disulfide crosslinking was performed by incubating ClpC (1 μM, wild type or cysteine variant), MecA (1.5 μM) and ClpP (2 μM, wild type or cysteine variant) in 50 mM HEPES pH 7.5, 25 mM KCl, 10 mM MgCl$_2$ in the presence of 2 mM ATPγS at 25 °C for 5 min. Crosslinking was started by the addition of copper-phenanthroline to a final concentration of 100 μM. Aliquots were taken and crosslinking was stopped by adding SDS sample buffer containing 4 mM iodacetamide (with or without β-mercaptoethanol). Samples were boiled and analyzed by SDS-PAGE followed by staining with SYPRO® Ruby Protein Gel Stain (ThermoFisher).

### Western blotting.
SDS-PAGEs were transferred to nitrocellulose or PVDF membranes by semi-dry blotting or wet blot transfer. Membranes were subsequently blocked with 3% BSA (w/v) in TBS-T. Custom-made MecA antibodies were used at 1:20,000 dilution. Anti-rabbit alkaline phosphatase conjugate (Vector Laboratories) was used as secondary antibody (1:10,000). Blots were developed using ECF™ Substrate (GE Healthcare) as reagent and imaged via Image-Reader LAS-4000 (Fujifilm).

### Sequence alignments
Sequence alignments were performed using ClustalW and displayed using Jalview (www.jalview.org) and Esprit for secondary structure superimposition (https://espript.ibcp.fr/ESPript/ESPript/index.php).

### Statistics and reproducibility
All the biochemical experiments report standard deviations based on at least three independent experiments. The cryo-EM raw data are deposited in the open database EMPIAR according to the FAIR principle.

**Table 1 | Cryo-EM data collection, refinement and validation statistics**

| Parameter | WT ClpC/ClpP body ClpC w/o NTD and MD (C1) EMDB: 51367 PDB: 9GI1 | WT MecA crown with ClpC NTD and MD (C6) EMDB: 51498 PDB: 9GOQ | WT clpP (D7) EMDB: 53538 PDB: 9R2S | WT MecA/ClpC/ClpP Composite map EMDB: 53879 PDB: 9RAI |
|---|---|---|---|---|
| Microscope | Titan Krios G3i (TFS) | Titan Krios G3i (TFS) | Titan Krios G3i (TFS) | Titan Krios G3i (TFS) |
| Detector and energy filter | Gatan K3 + Bioquantum(Ametek) | Gatan K3 + Bioquantum(Ametek) | Gatan K3 + Bioquantum(Ametek) | Gatan K3 + Bioquantum(Ametek) |
| Nominal magnification (nominal/calibrated at detector) | 105k | 105k | 105k | 105k |
| Voltage (kV) | 300 | 300 | 300 | 300 |
| Defocus range (µm) | −0.6 to −2.2 | −0.6 to −2.2 | −0.6 to −2.2 | −0.6 to −2.2 |
| Total electron exposure (or fluence, e-/Å$^2$) | 40 | 40 | 40 | 40 |
| Exposure rate (or flux, e-/pixel/s) | 15 | 15 | 15 | 15 |
| Number of frames collected | 40 | 40 | 40 | 40 |
| Pixel size (Å) | 0.828 (binned final 1.059) | 0.828 (binned final 1.059) | 0.828 (binned final 1.059) | 0.994 |
| Energy filter slit width (eV) | 20 | 20 | 20 | 20 |
| Automation software | EPU v3.7 | EPU v3.7 | EPU v3.7 | EPU v3.7 |
| # Micrographs used | 12,990 | 12,990 | 12,990 | 12,990 |
| Total # of extracted particles | 2,851,121 | 2,851,121 | 2,851,121 | 2,851,121 |
| Total # of refined particles (particles after removing junk) | 82,854 | 82,854 | 82,854 | 82,854 |
| # of particles in final map | 48,011 | 82,854 | 82,854 | 82,854 |
| Resolution of unmasked and masked reconstructions at 0.143 FSC | 4.1/2.9 | 4.0/3.4 | 2.8/2.4 | N/A |
| Local resolution range (Å) | 2–6 | 2.3–5.8 | 2.3–3.4 | N/A |
| Map sharpening B factor (Å$^2$)/(B factor Range) | −3 | −127 | −66 | N/A |
| Model composition | | | | |
| Non-hydrogen atoms | 47,097 | 13,878 | 19,981 | 60,975 |
| Protein residues | 6024 | 1704 | 2591 | 7728 |
| Ligands | MG:7, AGS:6;ADP:4 | 0 | 0 | MG:7, AGS:6;ADP:4 |
| *B* factors (Å$^2$) (min/max/mean) | | | | |
| Protein | 11.7/299.27/127.75 | 103.72/307.67/174.42 | 17.59/187.99/35.69 | 16.58/420.03/83.70 |
| Ligand | 79.98/179.82/108.55 | | | 37.85/119.65/74.77 |
| Map sharpening EMReady (any) | Yes | Yes | Yes | Yes |
| Atomic modeling refinement package(s) | Phenix 1.21 | Phenix 1.21 | Phenix 1.21 | Phenix 1.21 |
| | Coot 0.95 | Coot 0.95 | Coot 0.95 | Coot 0.95 |
| | Isolde | Isolde | Isolde | Isolde |
| CCvolume/CCmask | 0.80 | 0.87 | 0.89 | 0.90 |
| | 0/47,361 (0%) | 0/14,106 | 0/20,236 | 0/61,791 |
| Bad bond lengths and bad bond angles | 1/63,882 (0%) | 0/19,056 | 1/27,315 | 17/83,443 |
| Molprobity score | 1.41 | 1.23 | 1.24 | 1.89 |
| Clashscore | 3.52 | 1.85 | 2.22 | 10.67 |
| Ramachandran plot Z-score | −1.07 | −1.01 | −1 | 1.04 |
| Ramachandran Plot (%) | Outliers: 0.34% | Outliers: 0.18% | Outliers: 0.39% | Outliers: 0.04% |
| | Allowed: 2.94% | Allowed: 3.99% | Allowed: 2.51% | Favored: 96.94% |
| | Favored: 96.73% | Favored: 95.83% | Favored: 97.09% | |
| Ramachandran rotamers (%) | Outliers: 1.20% | Outliers: 0.80% | Outliers: 1.26% | Outliers: 1.64% |
| | Allowed: 5.51% | Allowed: 3.28% | Allowed: 4.93% | Favored: 76.22% |
| | Favored: 93.29% | Favored: 95.92% | Favored: 93.82% | |
| CaBLAM outliers (%) | 1.5% | 1.1%o | 1.8% | 1.6% |
| EMRinger score | 2.86 (unsharpen)/3.19 (EMReady) | 3.07 (symmetrized) | 4.66 (symmetrized) | N/A |

## Data availability

All cryo-EM density maps, half maps, masks, Fourier shell correlation curves and built models were deposited into the Electron Microscopy Data Bank (EMBD) (https://www.ebi.ac.uk/pdbe/emdb/) under accession codes EMD-51367 and pdb code 9GI1 for the ClpC/ClpP body, EMD-51498 and 9GOQ for the MecA crown, EMD-53538 and 9R2S for the ClpP body and EMD-53879 and pdb 9RAI for the composite map and model of the whole complex. Aligned micrographs and raw frames are available on EMPIAR with accession code EMPIAR-12887. The AlphaFold predictions and analysis are available on Figshare https://figshare.scilifelab.se/articles/dataset/Structure_of_the_central_Staphylococcus_aureus_AAA_protease_MecA_ClpC_ClpP/29261882. Source data for the figures is available in Supplementary Data 1.

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

## Acknowledgements

S.A. is supported by the Knut and Alice Wallenberg Foundation grant (KAW 2021-0347) to M.C. P.K. and T.J. were supported by the Heidelberg Biosciences International Graduate School (HBIGS). This work was funded by grants of the Knut and Alice Wallenberg Foundation (KAW 2021.0347) and Stiftelsen för Strategisk Forskning (RIF21-0047) to M.C. and of the Deutsche Forschungsgemeinschaft (MO 970/8-1) to A.M. The Cryo-EM facility at the Science for Life Laboratory Stockholm University (M.C., K.W.) is supported by grants from the Knut and Alice Wallenberg Foundation and the Family Erling Persson Foundation. We thank Stefan Fleischmann for IT support.

## Author contributions

S.A., K.W., P.K., A.M. and M.C. conceived the project and designed experiments. P.K., T.J. and A.M. performed protein isolation and biochemical characterization. S.A., K.W., and M.C. performed cryo-EM experiments and structural analysis. A.S. performed structural analysis. A.M. and M.C. wrote the manuscript with inputs from all authors.

## Funding

## Competing interests

The authors declare no competing interests.
