## [Transparent Peer Review file · Communications Biology]

Structure of the central *Staphylococcus aureus* AAA+ protease MecA/ClpC/ClpP

Corresponding Author: Dr Marta Carroni

Version 0:

Reviewer comments:

Reviewer #1

(Remarks to the Author)

This is a report on the manuscript "ClpC and ClpP act as reciprocal allosteric activators to form a highly efficient AAA+ protease" by Wallden et al.

In the above manuscript the authors describe the structure of the MecA-ClpC-ClpP complex from *S. aureus* as well as some additional *in vitro* experiments.

Overall, although the structural work appears to be carefully done in line with previous excellent publications by Carroni, the main findings described in the current manuscript are not particularly new or exciting. Reports from the Sauer group and more recently from the Goldberg/ Rubin have already demonstrated this kind of allosteric regulation between the 2 partners (protease and unfoldase). See for example (just for ClpC) Li et al "Structure and Functional Properties of the Active Form of the Proteolytic Complex, ClpP1P2, from *Mycobacterium tuberculosis*" or "Substrate delivery by the AAA+ ClpX and ClpC1 unfoldases activates the mycobacterial ClpP1P2 peptidase" by the Sauer group. In the first it is shown for example that binding of ClpC to ClpP increases the ClpC1 ATPase activity – to note that the authors here specifically discuss ATPase activation in the introduction ignoring this was already tested. Maybe the authors do not know these manuscripts, as they are not cited or referred, but the reciprocal activation between unfoldase (ClpC) and ClpP is not new and it is indeed already well established in the field. This said, we feel it is surely not appropriate to use the current title, that claims somehow the credit of this finding, and we believe that other title more focused on the structure determination is appropriate. As the authors know the present manuscript concerns mostly structural biology and in its core mostly supports previous observations.

Nevertheless, and despite the previous comments, we do believe that the manuscript is still important as it provides the first structure of the full complex ClpC-ClpP even at low resolution in some very important features (MecA ring).

Some important points below:

1) As stated above the following articles should be read and cited

"Substrate delivery by the AAA+ ClpX and ClpC1 unfoldases activates the mycobacterial ClpP1P2 peptidase"

"Structure and Functional Properties of the Active Form of the Proteolytic Complex, ClpP1P2, from *Mycobacterium tuberculosis*"

"Communication between ClpX and ClpP during substrate processing and degradation" likely the first article showing the reciprocal activation and modulation between the partners.

2)

The authors state that ClpP association reduces ATPase activity in ClpX in the introduction but previous results with MtbClpC (that does associate with MecA but is more similar than ClpX to SaClpC) have shown already that an ATPase activation is observed (see "Structure and Functional Properties of the Active Form of the Proteolytic Complex, ClpP1P2, from *Mycobacterium tuberculosis*") this should be cited and discussed.

3) We are aware that writing structural biology is a hard task but an effort should be made to make the article more approachable.

This is the first structure of a ClpC-ClpP complex – excluding the things we already know from other AAA+ proteins and are conserved in ClpC - what are the major differences observed (if any)? What is really new? We would suggest some rewriting to emphasize what is really new and make the manuscript less descriptive and therefore lighter for the reader.

For example: The authors test several beta hairpin ClpP mutants for peptidase activity (some are inactive) to later describe that several of the same mutants fail to form the 14 subunit ClpP – not surprising they also fail to function together with ClpC. As we introduce mutations in proteins folding or major structural modifications are not unexpected – it would be more simple to just state that these mutants fail to form the ClpP complex instead of testing them in other assays and show that they not work – what conclusions were the author expecting to obtain from these mutants if they do not form the ClpP barrel?

Reviewer #2

(Remarks to the Author)

In this manuscript, Wallden et al provide the cryoEM structure of the MecA/ClpC/ClpP complex from *Staphylococcus aureus*. ClpC is a AAA+ hexameric ATPase unfoldase, while ClpP is a cylindrical protease. MecA is considered to be an adaptor of ClpC. The authors find that MecA forms a dynamic crown on top of the ClpC/ClpP complex to regulate substrate access. The resolution for MecA is low in the structure. Most of the manuscript is spent on analyzing the interaction and allosteric regulation between ClpC and ClpP. The authors concentrate on carrying out mutational analysis on the P-loops, located at the bottom of the ClpC ATPase ring, that bind to hydrophobic pockets on the ClpP surface. They also concentrate on the N-terminal β -hairpins of ClpP that insert into the central ClpC threading channel and contact sites next to the ClpC ATPase center. For this complex, ClpC binding to ClpP shifts causes the protease to be more proteolytically active. Also, ClpP binding ClpC increases the ATPase activity of ClpC. These allosteric effects have been observed before for other AAA+-ClpP complexes. This study provides further insights into such allosteric regulations between the ClpP protease and its AAA+ cap.

The manuscript is well written, and the results are supported by the extensive mutational analysis data. I have a few comments.

1. I would recommend that the authors change the title of the manuscript as it is not very informative. They should mention in the title that they have a MecA-ClpC-ClpP structure. The issue of reciprocal allosteric regulation is already known for ClpP-AAA+ complexes.

2. More details are needed for the cryoEM data analysis. The data processing scheme needs to be shown. Also, did the authors only identify one state of the MecA/ClpC/ClpP complex?

3. Figure 2a, label the subunits of ClpP.

4. Page 10 – The use of ADEP1 to propose that the ClpC P-loops undergo rapid cycling is not founded. What the experiments demonstrate is that binding of ADEP1 to ClpP can out compete the binding of ClpC to ClpP. How fast ClpC cycles in its binding to ClpP is not relevant in these experiments. To make a clearer conclusion, the authors need to measure K_d 's and k_{on} and k_{off} rates. The authors should reword this section or do more experiments.

5. Provide a better view in the respective Figure for this description:

“The ClpP N-terminal β -hairpins contact the ClpC inner channel more on one side of the ring (ClpC protomers f-c (Fig. 2a/e).”

6. Given that the binding of SaClpP to SaClpC enhances the ATPase activity of SaClpC, while the opposite is found for the binding of EcClpP to EcClpX, it would be important for the authors to carry out a detailed analysis of these observations using their cryoEM structure of SaClpCP-MecA and the published cryoEM structures of EcClpXP.

Reviewer #3

(Remarks to the Author)

This study from Wallden et al. presents the cryoEM structure of *S. aureus* MecA/ClpC/ClpP complex, and probes the functional importance of ClpC P-loops and the ClpP N-terminal beta-hairpin in mediating the ClpP-ClpC interaction. The authors carry out complementary structural and biochemical studies to establish the functional importance of these interactions. Notably, association between ClpC and ClpP increases the unfolding activity of ClpC, while at the same time enhancing peptidase activity of ClpP. The ClpC-ClpP structure includes the adaptor protein MecA, but the moderate resolution of the MecA (6-9 Å) precludes detailed mechanistic insights into adaptor-protein binding and substrate recruitment. Given that these proteins (or homologs) have been extensively characterized, many of the findings serve to support the conclusions of prior studies. The novelty of the study stems from extensive analysis of the interactions between the ClpC P-loop and ClpP beta-hairpin aimed at better understanding the putative allosteric communication between the ATPase and the peptidase. Insights are mainly drawn from mutagenesis on the residues contained in the ClpP beta-hairpin (I8 to R16). The authors find that most of these mutations disrupt the oligomerization of ClpP and its peptidase activities, as

well as disrupt the capacity for ClpP to form a complex with MecA/ClpC. A complementary experiment showed that ADEP1 binding compensates for defective peptidase activities of most of mutant ClpP. While the functional studies were performed rigorously, the authors don't fully relate the mutagenesis studies back to the structure as a means of detailing why certain residues and their interactions may be more important than others, or why certain mutations would perturb ClpP assembly. Based on the title and abstract, I was particularly excited to learn how the ClpC and ClpP "function as reciprocal allosteric regulators," but the manuscript falls short of providing molecular insights beyond the finding that the ClpP beta-hairpin is important for function. How is the mechanism of reciprocity? There is a strong foundation for a very interesting study in these data, and with a bit more experimentation and thoughtful analysis/integration of the findings into a cogent molecular mechanism, this would be an impactful body of work. However, the study in its current form constitutes what I consider to be an incremental advance in the field, and it is not clear that the impact of the findings is aligned with the publication goals of Communications Biology.

Major issues:

- The manuscript would significantly be improved by providing a detailed assessment of how mutations may relate to the structure. Given that the authors have obtained an atomic model of the ClpP-ClpC interactions, they should be able to relate the mutagenesis results to the structural interactions, and provide insights into how mutations would disrupt complex formation and function. In particular, while the authors showed that ClpP enhances the activities of its AAA+ partners, the authors don't propose a mechanism by which this modulation of the activities. Further insights into an allosteric regulatory function could be gained through mutations in regions in alloR and alloW that contact the beta-hairpins. Overall, the authors state what they observe without providing mechanistic relevance that is tied to the functional and biochemical studies.
- One finding that warrants further investigation is the ADEP1-mediated restoration of peptidase activity in mutant ClpP. This is particularly interesting, given that the majority of these mutants don't form a functional oligomer. Is ADEP1 mediating inter-ClpP interactions to enable oligomer formation? How could such a mechanism function? Structural insights into the ADEP1 restoration of ClpP activity would vastly increase the impact of this finding.
- While there are numerous comparisons to other ClpP-related systems throughout the manuscript, the authors don't detail these comparisons in accompanying figures – this would be particularly useful for structural details that the authors identify to be unique to ClpP-ClpC.
- In the discussion, authors propose a "sensing" function, as opposed to a direct stimulating effect of ClpP beta-hairpin on ATP hydrolysis. This proposal seems incongruent with the findings that ClpP binding to ClpC stimulates its ATPase activities. This should be reconsidered or the proposed mechanism further clarified in light of the findings.
- The introduction is a bit long-winded covering aspects that are not relevant to the major findings of the paper. This could be condensed while still being approachable to non-experts in ClpP-ClpC interactions. The discussion section mostly reiterates findings already discussed in the manuscripts, as opposed to a discussion that integrates the findings into a cohesive allosteric mechanism that constitutes an advance in our understanding of this important complex.
- The quality of the EM density for the nucleotides is difficult to assess, as the authors have "zoned" the cryoEM map to only display voxels within a certain radius of the nucleotide to show that EM density is present for the model. Low contour values can be used to produce density surrounding any modeled atom within a map. While I do not suspect that the authors are being intentionally disingenuous in their analysis, the density for surrounding residues (particularly Arginine fingers) should be included in this figure or in a supplementary figure depicting this region. Based on Figure 4a, the density for the nucleotides doesn't seem of sufficient quality to unambiguously assign nucleotide state.

Importantly, the EM methods and validation is lacking in the manuscript, which makes overall assessment of the quality of the models challenging. Please address the following:

In the methods, the authors should include:

- device and settings used for glow discharge of EM grids
- type of blotting paper used for EM grid preparation
- Vitrobot settings used for blotting and vitrification
- all non-default parameters used for classification and refinement (mask diameters, tau-fudge, e-step, # of classes, initial low pass filter, # of iterations, etc.)
- a more detailed description of the atomic modeling methodology, including any relevant Phenix refinement parameters and any constraints (secondary structure, Ramachandran, etc.)

The supplement should include:

- A processing workflow figure
- FSCs (preferably 3D FSCs output from 3dfsc.salk.edu), with FSC at 0.143 denoted
- Model-to-Map FSC(s), with resolution at 0.5 FSC cutoff denoted
- Figure 1 (local resolution map) can be moved the supplement, this panel should focus more on the domain/subunit organization of the complex

For straightforward reference and readability, please include in a supplemental Table:

- EMDB & PDB IDs
- Microscope
- Detector

- Magnification (nominal/calibrated at detector)
- Voltage (kV)
- Defocus range (um)
- Total electron exposure (or fluence, e-/Å²)
- Exposure rate (or flux, e-/pixel/s)
- Number of frames collected
- Pixel size (Å)
- Energy filter slit width (eV)
- Automation software (EPU, SerialEM, Legion, etc.)
- # Micrographs used
- Total # of extracted particles
- Total # of refined particles (particles after removing junk)
- # of particles in final map
- Estimated error of translations/rotations
- Resolution of unmasked and masked reconstructions at 0.5 and 0.143 FSC
- Local resolution range
- 3DFSC Sphericity value
- Map sharpening B factor (Å²) / (B factor Range)
- Model composition (protein, ligands, DNA/RNA)
- Atomic modeling refinement package(s)
- CCvolume/CCmask
- B factors of protein residues & ligands
- R.m.s. deviations from ideal values (bond lengths & bond angles)
- Molprobity score
- Clashscore
- Poor rotamers (%)
- Ramachandrans (Favored, Outliers (%))
- CaBLAM outliers (%)
- EMRinger score

I do not review anonymously, and encourage the authors to publicly share their submitted manuscript on a preprint server (e.g. bioRxiv) at this time. This practice enables others to consider the findings presented in this research, as well as providing the authors with feedback from the community alongside formal peer review. Importantly, this practice is allowed (but not required) by Communications Biology. A graduate in my lab, Wenqian “Kelly” Chen, helped with this review.

-Gabe Lander

Version 2:

Reviewer comments:

Reviewer #1

(Remarks to the Author)

The authors have clearly made a considerable effort to improve the manuscript, which I view positively. They have also addressed comments from the various reviewers.

I had no major concerns regarding the structural data. The structures of AAA+ unfoldases in complex with their partner proteases are inherently challenging to obtain, and in my view, the quality of the structural data alone would justify the publication of this work. In this revised version, the authors further improve the structural component and also attempt to expand the associated biochemical analysis. Again, I find no issue with the structural work — as is often the case from the Carroni group, it appears to be carefully and thoroughly executed.

However, I find some of the new biochemical data, particularly the mixing experiments presented in Fig. S8, difficult to interpret. I do not understand what the authors intend to demonstrate with these assays. First, the term "concentration-sensitive range" is unclear and should be defined more precisely. Additionally, what conclusions can truly be drawn from mixing WT ClpC with mutant ClpC? Do they form mixed hexamers? Are the association constants for WT and mutant ClpC the same? None of these foundational assumptions are addressed or experimentally demonstrated. Furthermore, the use of buffer as a control is not ideal. A better control would be to use an unrelated protein at similar concentration, or to test the effect of adding increasing amounts of WT protein and observing whether ATPase activity increases proportionally.

In addition, the statement that 4–5 subunits need to be active for ClpC function is not convincingly supported by the current data. To rigorously test this, the authors would need to adopt an approach like that used by the Sauer group for ClpX, involving covalently linked hexamers. This allows specific subunits to be mutated while keeping the hexamer intact, making it possible to test how many inactive subunits are compatible with activity. I understand that constructing covalently linked ClpC hexamers is technically challenging — and may not even yield a soluble or functional complex — but in light of this, I

would recommend removing the current mixing experiments from the manuscript, as they are not informative and may detract from the overall clarity.

Lastly, I want to reiterate a point from my previous review: the continued use of ClpP mutants that fail to assemble properly is problematic. While I appreciate the effort required to generate and purify these mutants, if a ClpP variant does not form a complex, any downstream functional assays involving ClpC will be inherently compromised and hard to discuss. Including data from such mutants across multiple assays does not add value and only makes the manuscript harder to read and interpret. Since this issue has already been noted, and no new experiments are needed, I would suggest simply removing or minimising the use of those non-assembling mutants to improve clarity - just focus on the other ones.

Reviewer #2

(Remarks to the Author)

The authors have addressed my comments as best as they can. I do not have further comments.

Reviewer #3

(Remarks to the Author)

I appreciate the efforts the authors put into addressing our concerns, and believe that the manuscript is much improved. I am satisfied with the responses and find the manuscript appropriate for publication.

Version 3:

Reviewer comments:

Reviewer #1

(Remarks to the Author)

The article has been improved once again — while there is always room for further refinement, I believe it has now reached a state suitable for publication

Answer to reviewers about the manuscript

Structure of the central *Staphylococcus aureus* AAA+ protease MecA/ClpC/ClpP

Comments of the reviewers are shown in blue italics.

General note to all reviewers:

We thank the reviewers for their valuable comments. We have taken our time to extensively revisit our whole work. We have collected and analysed new cryo-EM datasets of the MecA/ClpC/ClpP protease in complex with its natural substrate FtsZ with more modern electron detectors and computational methods. This resulted in overall better-quality maps that allow us to propose a possible mechanism, (i) how MecA can deliver bound substrate to the ClpC/ClpP protease and (ii) how ClpP association enhances ATPase and threading activities of ClpC. These mechanistic findings could not be derived from the cryoEM structure submitted before. We additionally complemented the new structural work with biochemical analyses of ClpC mutants and mixed ClpC oligomeric assemblies. Our new findings led to a complete revision of our original work with almost all figures renewed.

According to the suggestions of the reviewers we have changed the title of the manuscript to "**Structure of the central *Staphylococcus aureus* AAA+ protease MecA/ClpC/ClpP**". Furthermore, we revised introduction and discussion according to the reviewer's advice and referred to the indicated as well as newly published work.

Reviewer 1

Overall, although the structural work appears to be carefully done in line with previous excellent publications by Carroni, the main findings described in the current manuscript are not particularly new or exciting. Reports from the Sauer group and more recently from the Goldberg/ Rubin have already demonstrated this kind of allosteric regulation between the 2 partners (protease and unfoldase). See for example (just for ClpC) Li et al "Structure and Functional Properties of the Active Form of the Proteolytic Complex, ClpP1P2, from Mycobacterium tuberculosis" or " Substrate delivery by the AAA+ ClpX and ClpC1 unfoldases activates the mycobacterial ClpP1P2 peptidase" by the Sauer group. In the first it is shown for example that binding of ClpC to ClpP increases the ClpC1 ATPase activity – to note that the authors here specifically discuss ATPase activation in the introduction ignoring this was already tested. Maybe the authors do not know these manuscripts, as they are not cited or referred, but the reciprocal activation between unfoldase (ClpC) and ClpP is not new and it is indeed already well established in the field. This said, we felt it is surely not appropriate to use the current title, that claims somehow the credit of this finding, and we believe that other title more focused on the structure determination is appropriate. As the authors know the present manuscript concerns mostly structural biology and in its core mostly supports previous observations.

We thank the reviewer for their valuable comments. We have changed the title of the manuscript to "**Structure of the central *Staphylococcus aureus* AAA+ protease MecA/ClpC/ClpP**". We have revised the introduction and the discussion according to the advice and referred to the indicated published work as well as to the most recent work on the subject that was published or put in pre-print lately.

*1)As stated above the following articles should be read and cited
"Substrate delivery by the AAA+ ClpX and ClpC1 unfoldases activates the mycobacterial ClpP1P2 peptidase"*

This work (Schmitz and Sauer 2014) is now cited and discussed as it shows the allosteric stabilizing effect of the AAA+ partner over the ClpP1P2 peptidase in *Mycobacterium*.

“Structure and Functional Properties of the Active Form of the Proteolytic Complex, ClpP1P2, from Mycobacterium tuberculosis”

This work (M. Li et al. 2016) is now cited and discussed as it shows the allosteric activating effect of the Bz-LL-activated ClpP1P2 over the ClpC1 ATPase in *Mycobacterium*. This is in good agreement with the *S. aureus* case presented in our study.

“Communication between ClpX and ClpP during substrate processing and degradation” likely the first article showing the reciprocal activation and modulation between the partners.

This work (Joshi et al. 2004) is now cited and discussed. It shows that in *E. coli* the peptidase ClpP (WT) reduces the ATPase and unfolding activity of ClpX-WT, while it rescues ATPase defects in ClpX mutants at the same time. Notably, different observations have been reported for *Mycobacterium tuberculosis*, where ClpP1P2 association enhances ClpX ATPase activity (Li et al. 2016). The mechanistic reasons underlying the differing effects of peptidase association on ClpX ATPase activity are unclear.

2) The authors state that ClpP association reduces ATPase activity in ClpX in the introduction but previous results with MtbClpC (that does associate with MecA but is more similar than ClpX to SaClpC) have shown already that an ATPase activation is observed (see “Structure and Functional Properties of the Active Form of the Proteolytic Complex, ClpP1P2, from Mycobacterium tuberculosis”) this should be cited and discussed.

We have added the respective work (Li et al. 2016) to the manuscript. These findings pinpoint similarities in the cooperations between Sa ClpC and Mtb ClpC1 and their respective ClpP and ClpP1/ClpP2 peptidases and underline the role of the protease partner in AAA+ protein activation.

3) We are aware that writing structural biology is a hard task but an effort should be made to make the article more approachable.

This is the first structure of a ClpC-ClpP complex – excluding the things we already know from other AAA+ proteins and are conserved in ClpC - what are the major differences observed (if any)? What is really new? We would suggest some rewriting to emphasize what is really new and make the manuscript less descriptive and therefore lighter for the reader.

As we have mostly re-written the manuscript adding new data, we hope to have managed making the reading smoother. Figures are also mostly new and hopefully they convey the main pieces of information that we can gather from our data: (i) an analysis of the MecA part of the complex and its motion (new Fig. 2) and (ii) a possible rational for the activation of the ClpC ATPase activity via ClpP (new Fig.6).

For example: The authors test several beta hairpin ClpP mutants for peptidase activity (some are inactive) to later describe that several of the same mutants fail to form the 14 subunit ClpP – not surprising they also fail to function together with ClpC. As we introduce mutations in proteins folding or major structural modifications are not unexpected – it would be more simple to just state that these mutants fail to form the ClpP complex instead of testing them in other assays and show that they not work – what conclusions were the author expecting to obtain from these mutants if they do not form the ClpP barrel?

We followed the suggestion of the reviewer and shortened repetitive parts of the manuscript and data discussion of the ClpP mutants that exhibit oligomerization defects.

Reviewer #2

1. *I would recommend that the authors change the title of the manuscript as it is not very informative. They should mention in the title that they have a MecA-ClpC-ClpP structure. The issue of reciprocal allosteric regulation is already known for ClpP-AAA+ complexes.*

We agree with the reviewer and it was also pointed out by reviewer #1. We have changed the title to: Structure of the central *Staphylococcus aureus* AAA+ protease MecA/ClpC/ClpP”.

2. *More details are needed for the cryoEM data analysis. The data processing scheme needs to be shown. Also, did the authors only identify one state of the MecA/ClpC/ClpP complex?*

We agree with the reviewer and for this completely new dataset and data analysis we have reported the processing scheme (new Supplementary Fig. 1). Same as for the old dataset, we only observe one state of the MecA/ClpC/ClpP complex even upon extensive classification, both global and focused. We have also a ClpC double Walker B (ATPase deficient mutant) dataset (in complex with MecA and ClpP) that also only shows one state. This is different from most other AAA+/ClpP complex structures determined by cryo-EM to date. We reason that this could be caused by the presence of the MecA crown that partially restrains the ClpC movements or makes them more synchronised. We are depositing the raw data in EMPIAR so that they will be available for anyone testing new analysis method and dig possible rare conformations.

3. *Figure 2a, label the subunits of ClpP.*

The new manuscript contains completely new Figures. Hopefully, labelling is better and more consistent. The labelling (subunits a-f for ClpC and Pa-Pn for the ClpP protomers) match the chain ID of the deposited pdb models.

4. *Page 10 – The use of ADEP1 to propose that the ClpC P-loops undergo rapid cycling is not founded. What the experiments demonstrate is that binding of ADEP1 to ClpP can out compete the binding of ClpC to ClpP. How fast ClpC cycles in its binding to ClpP is not relevant in these experiments. To make a clearer conclusion, the authors need to measure Kd's and kon and koff rates. The authors should reword this section or do more experiments.*

We agree with the concern of the reviewer and have removed the respective data from the manuscript.

5. *Provide a better view in the respective Figure for this description: “The ClpP N-terminal b-hairpins contact the ClpC inner channel more on one side of the ring (ClpC protomers f-c (Fig. 2a/e).”*

The new manuscript contains a better refined map and completely new figures. We can now show the specific contacts of each ClpC subunits to each ClpP protomer β -hairpin (new Fig. 5e).

6. *Given that the binding of SaClpP to SaClpC enhances the ATPase activity of SaClpC, while the opposite is found for the binding of EcClpP to EcClpX, it would be important*

for the authors to carry out a detailed analysis of these observations using their cryoEM structure of SaClpCP-MecA and the published cryoEM structures of EcClpXP.

The inhibitory effects of *E.coli* ClpP on ClpX ATPase activity is specific as activation of *M. tuberculosis* ClpX ATPase activity upon association with ClpP1P2 has been described, as also pointed out by reviewer 1 (see above) (M. Li et al. 2016). Furthermore, *E. coli* ClpP can increase ATPase activity of ClpX mutants (Joshi et al. 2004). Thus, the impact of *E. coli* ClpP on ClpX activity is context dependent. The available data therefore do not support major mechanistic differences in the cooperation between ClpC/ClpX with ClpP. Accordingly, the contact sites between *Neisseria meningitidis* ClpX (Ripstein et al. 2020) and ClpP and the ones described in this study are similar, including the interaction between ClpP β -hairpins and the allo-W region of the AAA+ partner. We added a respective statement to the manuscript.

As it had been also required by reviewer #3, we have looked at the published *E. coli* ClpX/ClpP cryo-EM structures (Fei et al. 2020) in the regions of interaction (allo-W, allo-R and P-loops) at the interface between a AAA-2 subunit with ADP and one with ATP bound (see Figure 1 below). The overall pattern of interactions, between the β -hairpins and the P-loops and between the β -hairpins and the allo-R region preceding the R-finger, are maintained. However, there are major differences between *E. coli* ClpX and *S. aureus* ClpC, as expected by the low sequence identity (see alignment below). The P-loop in ClpX is longer, but, as in ClpC, it does get contacted by the ClpP β -hairpin in the ATP bound subunit. The allo-R region is made of three helices instead of two, but, as in ClpC, it does get contacted by ClpP β -hairpin in the ADP-bound subunit. The allo-W residue in *E.coli* ClpX (D187) is not in contact with the ClpP β -hairpins. In *N. meningitidis* ClpX/ClpP the ClpP β -hairpin has been reported to contact the ClpX pore-loop 2 that correspond to the allo-W. It is thus very difficult to draw any conclusion about specific mechanisms of activation or reduction of the ATPase activity of ClpX by ClpP. It should also be noted that the overall conservation between ClpC and ClpX in the ClpP-interacting regions is not high (see Supplementary Fig. 5a and Figure 2), further complicating the direct comparison of ClpX vs ClpC cooperations with ClpP. We therefore prefer not adding such comparison to the manuscript. We believe this is also outside the scope of this study.

E.coli ClpX/ClpP pdb code 6p01, subunit E-ATP and F-ADP:

Figure 1 Visualisation and analysis of the interface between an ADP and an ATP bound subunit in *E.coli* ClpX from the ClpX/ClpP complex (pdb 6p01, (Fei et al. 2020)). This interface and contacts are the same in *N. meningitidis* ClpX/ClpP configurationA (Ripstein et al. 2020). Overall the pattern of interactions between the ClpP β -hairpins and the ClpX regions adjacent to the ATP pockets (allo-R in magenta) and to the ClpX P-loops (P-loop in green) are similar to the *S.aureus* ClpC/ClpP. However, compared to *S.aureus* ClpC/ClpP where two adjacent ClpP β -hairpins contact two adjacent ClpC subunits, in *E.coli* ClpX there is in the middle a ClpP β -hairpin that doesn't contact any ClpX subunit.

Figure 2: **Sequence alignment** of the ClpP-interacting AAA domain of ClpC, ClpA and ClpX from *Staphylococcus aureus*, *Bacillus subtilis* and *Escherichia coli*. Postions of Walker A and B motifs (red) and ClpP-interacting parts (pore-2 loop/allo-W, P-loop, allo-R) of ClpC, ClpA and ClpX are indicated.

A comparison to ClpA is provided in the answer to reviewer #3 below.

Reviewer #3

1. *The manuscript would significantly be improved by providing a detailed assessment of how mutations may relate to the structure. Given that the authors have obtained an atomic model of the ClpP-ClpC interactions, they should be able to relate the mutagenesis results to the structural interactions, and provide insights into how mutations would disrupt complex formation and function. In particular, while the authors showed that ClpP enhances the activities of its AAA+ partners, the authors don't propose a mechanism by which this modulation of the activities. Further insights into an allosteric regulatory function could be gained through mutations in regions in alloR and alloW that contact the beta-hairpins. Overall, the authors state what they observe without providing mechanistic relevance that is tied to the functional and biochemical studies.*

Using the new structure, we can now describe more accurately the contacts between the β -hairpins from different ClpP protomers and the ClpC subunits (new Fig. 5). This also allows us to explain phenotypes of ClpP β -hairpin mutants. We additionally propose a possible mechanism, described in the revised manuscript and in the new Fig. 6, which implies a readjustment of the ATP pocket and of the R-finger at the interface between a AAA2 domain that has undergone hydrolysis and its neighbour,

which could/should hydrolyse next. This repositioning of the ATPase pockets is facilitated by contacts mediated by the allo-W and allo-R regions. So, we followed the suggestion of the reviewer and studied the consequences of ClpC alloW (E620A/K621A) and alloR (E692A/K694A) mutants on activation by ClpP. Additionally, we analyzed the role of the conserved N659 residue, which our structural analysis implicates in signaling between ClpP docking sites and the AAA2 ATPase center. We found that the alloW and N659A mutants have reduced degradation activity (new Fig.7 and Supplementary Fig.10), which could be linked to reduced ATPase activity. This can be rationalized by close vicinity of the alloW region and N659 to the Walker B motif and bound nucleotide and thus the catalytic site of the AAA2 domain. Importantly, ClpC-E620A/K621A and ClpC-N659A ATPase activities were no longer stimulated in presence of ClpP (new Fig. 7e). Similarly, unfolding activities of ClpC-E620A/K621A and ClpC-N659A were much less enhanced upon ClpP association as compared to ClpC-WT (new Fig. 7f, Supplementary Fig.10d-e). In contrast, the alloR mutant E692A/K694A showed WT-like degradation activities. Notably, this mutant showed a partially reduced ATPase activity that was restored to WT-like activities upon ClpP association. This suggests that ClpP binding compensates for the partial defects caused by the allo-R mutation. We conclude that ClpP contacts with the allo-W region are crucial for ATPase activation. Contacts between ClpP and allo-R are instead more complex and point on a potential role of fine tuning ClpC-ClpP cooperation.

2. *One finding that warrants further investigation is the ADEP1-mediated restoration of peptidase activity in mutant ClpP. This is particularly interesting, given that the majority of these mutants don't form a functional oligomer. Is ADEP1 mediating inter-ClpP interactions to enable oligomer formation? How could such a mechanism function? Structural insights into the ADEP1 restoration of ClpP activity would vastly increase the impact of this finding.*

ClpP represents a highly allosteric system, in which binding of e.g. ADEP1 induces long-range conformational changes including the N-terminal β -hairpins and interactions between ClpP monomers and heptameric rings. Similarly, mutations in the N-terminal loops have been shown to affect structural integrity of the *S. aureus* ClpP barrel, underlining allosteric communications between diverse ClpP sites (3). Structural defects of *S. aureus* N-terminal loop mutants can be rescued by ADEP binding (Vahidi et al. 2018). We consider this observation very similar to ours: the partial rescue of N-terminal loop mutants upon ADEP binding. ADEP binding leads to ordering of N-terminal loops into β -hairpin structures (D. H. S. Li et al. 2010) and this effect likely restores proper β -hairpin formation in our β -hairpin mutants. Notably, ADEP binding is crucial for oligomerization of *B. subtilis* and human ClpP (Kirstein et al. 2009; Lowth et al. 2012), underlining the impact of the compound on ClpP ring formation and stability. We have added these clarifications to the revised manuscript.

3. *While there are numerous comparisons to other ClpP-related systems throughout the manuscript, the authors don't detail these comparisons in accompanying figures – this would be particularly useful for structural details that the authors identify to be unique to ClpP-ClpC.*

We have partially answered this question to reviewer #2, regarding comparison to *E.coli* ClpX/ClpP (Fei et al. 2020). In this new amply revised manuscript, we have a more complete analysis of each ClpC subunit contact with each ClpP protomer β -hairpin and we refer to the relevant similarities for other AAA+/ClpP systems. The overall pattern of interaction and the residues that we have identified as important in the allo-W and allo-R regions are conserved in the Hsp100 proteins ClpC, ClpA and ClpE (new Supplementary Fig. 5a and Figure 2 above). We do not endeavour in a comparison of each and every subunit of all the existing Hsp100s in complex with ClpP. Such a detailed analysis could be the subject of a review, but ideally it

should be supported by mutational analysis in the different AAA+ protease systems. Such mutational analysis is provided here for ClpC/ClpP but is missing for the other AAA+ proteases.

We also looked at the *E. coli* ClpA/ClpP structures for the subunits at the interface between the inactive ADP-bound part of the ring and the ATP-bound active adjacent subunit. The positions of the allosteric regions (allo-W, allo-R and P-loops) are conserved in ClpA relative to the ClpP protomer β -hairpins underneath (see Figure 3 below). When we overlap the subunits at the inactive/active interfaces from different configurations of the ClpA/ClpP (Lopez et al. 2020) with the one from *S. aureus* ClpC/ClpP we see an almost perfect overlap of all domains, but a change in the relative positions of the other AAA+ subunits and more interestingly of the ClpP barrel. The mechanistic relevance of these structural differences is, however, unclear as *E. coli* ClpP association also enhances ATPase activity of ClpA. We therefore prefer not adding such comparison to the manuscript. We also believe that such analysis will be outside the scope of this study.

Figure 3 Superimposition of two subunits making the interface between the inactive, ADP-bound, and active ATP-bound part of the AAA-2 ring in the *S.aureus* ClpC/ClpP complex (this study) and *E.coli* ClpA/ClpP complex. The high sequence and structural conservation of the P-loop, allo-R and allo-W regions suggest that the allosteric effect of ClpP onto the AAA+ partner and its mechanism can be conserved in Hsp100 unfoldases.

- In the discussion, authors propose a “sensing” function, as opposed to a direct stimulating effect of ClpP beta-hairpin on ATP hydrolysis. This proposal seems incongruent with the findings that ClpP binding to ClpC stimulates its ATPase activities. This should be reconsidered or the proposed mechanism further clarified in light of the findings.*

As explained in the answer to the first question above, we now propose a possible way of ClpP facilitating sequential and more efficient ATP hydrolysis around the ClpC ring. The model is partially validated by ClpC mutants (e.g. allo-W E620A/K621A), which are no longer stimulated by ClpP association (new Fig 7e). Since the allo-R mutant E692A/K694A is still activated by ClpP, we speculate that the allosteric activation of ClpP onto ClpC is far more intricate and tunable by a number of residues, than the simple activation path that we propose (new Fig. 6). In an attempt to propose a detailed mechanism, we started molecular dynamics simulations of the complex. As these calculations are fairly expensive, we plan to release the results in an upcoming publication.

5. *The introduction is a bit long-winded covering aspects that are not relevant to the major findings of the paper. This could be condensed while still being approachable to non-experts in ClpP-ClpC interactions. The discussion section mostly reiterates findings already discussed in the manuscripts, as opposed to a discussion that integrates the findings into a cohesive allosteric mechanism that constitutes an advance in our understanding of this important complex.*

We followed the suggestion of the reviewer and completely revised introduction and discussion accordingly also in light of new results published by other groups in the meantime.

6. *-The quality of the EM density for the nucleotides is difficult to assess, as the authors have "zoned" the cryoEM map to only display voxels within a certain radius of the nucleotide to show that EM density is present for the model. Low contour values can be used to produce density surrounding any modeled atom within a map. While I do not suspect that the authors are being intentionally disingenuous in their analysis, the density for surrounding residues (particularly Arginine fingers) should be included in this figure or in a supplementary figure depicting this region. Based on Figure 4a, the density for the nucleotides doesn't seem of sufficient quality to unambiguously assign nucleotide state.*

In this completely revised manuscript, we present new structural data and we show all the density for the nucleotides and surrounding areas (new Fig. 6 a). We also report in the text and caption about any doubtful interpretation of the density, such as in the case of the ATP pocket of subunit a AAA2. Built models and maps (both sharpened and unsharpened) are available for the reviewers in a shared folder provided to the editor (link).

7. *Importantly, the EM methods and validation is lacking in the manuscript, which makes overall assessment of the quality of the models challenging.*

We have addressed all points raised by the reviewer and added respective information to the methods section and in the statistics table. For completeness we also make the models and maps available and we are depositing the raw data to EMPIAR.

References

- Fei, Xue, Tristan A. Bell, Simon Jenni, Benjamin M. Stinson, Tania A. Baker, Stephen C. Harrison, and Robert T. Sauer. 2020. "Structures of the ATP-Fueled ClpXP Proteolytic Machine Bound to Protein Substrate." *ELife* 9 (February). <https://doi.org/10.7554/eLife.52774>.
- Joshi, Shilpa A., Greg L. Hersch, Tania A. Baker, and Robert T. Sauer. 2004. "Communication between ClpX and ClpP during Substrate Processing and Degradation." *Nature Structural & Molecular Biology* 11 (5): 404–11.
- Kirstein, Janine, Anja Hoffmann, Hauke Lilie, Ronny Schmidt, Helga Rübsamen-Waigmann, Heike Brötz-Oesterhelt, Axel Mogk, and Kürşad Turgay. 2009. "The Antibiotic ADEP Reprogrammes ClpP, Switching It from a Regulated to an Uncontrolled Protease." *EMBO Molecular Medicine* 1 (1): 37–49.
- Li, Dominic Him Shun, Yu Seon Chung, Melanie Gloyd, Ebenezer Joseph, Rodolfo Ghirlando, Gerard D. Wright, Yi-Qiang Cheng, Michael R. Maurizi, Alba Guarné, and Joaquin Ortega. 2010. "Acyldepsipeptide Antibiotics Induce the Formation of a Structured Axial Channel in ClpP: A Model for the ClpX/ClpA-Bound State of ClpP." *Chemistry & Biology* 17 (9): 959–69.
- Li, Mi, Olga Kandror, Tatos Akopian, Poorva Dharkar, Alexander Wlodawer, Michael R. Maurizi, and Alfred L. Goldberg. 2016. "Structure and Functional Properties of the Active Form of the Proteolytic Complex, ClpP1P2, from Mycobacterium Tuberculosis." *The Journal of Biological Chemistry* 291 (14): 7465–76.

- Lopez, Kyle E., Alexandra N. Rizo, Eric Tse, Jiabei Lin, Nathaniel W. Scull, Aye C. Thwin, Aaron L. Lucius, James Shorter, and Daniel R. Southworth. 2020. "Conformational Plasticity of the ClpAP AAA+ Protease Couples Protein Unfolding and Proteolysis." *Nature Structural & Molecular Biology* 27 (5): 406–16.
- Lowth, Bradley R., Janine Kirstein-Miles, Tamanna Saiyed, Heike Brötz-Oesterhelt, Richard I. Morimoto, Kaye N. Truscott, and David A. Dougan. 2012. "Substrate Recognition and Processing by a Walker B Mutant of the Human Mitochondrial AAA+ Protein CLPX." *Journal of Structural Biology* 179 (2): 193–201.
- Ripstein, Zev A., Siavash Vahidi, Walid A. Houry, John L. Rubinstein, and Lewis E. Kay. 2020. "A Processive Rotary Mechanism Couples Substrate Unfolding and Proteolysis in the ClpXP Degradation Machinery." *ELife* 9 (January). <https://doi.org/10.7554/eLife.52158>.
- Schmitz, Karl R., and Robert T. Sauer. 2014. "Substrate Delivery by the AAA+ ClpX and ClpC1 Unfoldases Activates the Mycobacterial ClpP1P2 Peptidase: Substrate Activation of Mycobacterial ClpP." *Molecular Microbiology* 93 (4): 617–28.
- Vahidi, Siavash, Zev A. Ripstein, Massimiliano Bonomi, Tairan Yuwen, Mark F. Mabanglo, Jordan B. Juravsky, Kamran Rizzolo, et al. 2018. "Reversible Inhibition of the ClpP Protease via an N-Terminal Conformational Switch." *Proceedings of the National Academy of Sciences of the United States of America* 115 (28): E6447–56.

Answer to reviewers about the manuscript

Structure of the central *Staphylococcus aureus* AAA+ protease MecA/ClpC/ClpP

Comments of the reviewers are shown in blue italics.

General note to all reviewers:

We thank the reviewers for their valuable comments. We have taken our time to extensively revisit our whole work. We have collected and analysed new cryo-EM datasets of the MecA/ClpC/ClpP protease in complex with its natural substrate FtsZ with more modern electron detectors and computational methods. This resulted in overall better-quality maps that allow us to propose a possible mechanism, (i) how MecA can deliver bound substrate to the ClpC/ClpP protease and (ii) how ClpP association enhances ATPase and threading activities of ClpC. These mechanistic findings could not be derived from the cryoEM structure submitted before. We additionally complemented the new structural work with biochemical analyses of ClpC mutants and mixed ClpC oligomeric assemblies. Our new findings led to a complete revision of our original work with almost all figures renewed.

According to the suggestions of the reviewers we have changed the title of the manuscript to "**Structure of the central *Staphylococcus aureus* AAA+ protease MecA/ClpC/ClpP**". Furthermore, we revised introduction and discussion according to the reviewer's advice and referred to the indicated as well as newly published work.

Reviewer 1

Overall, although the structural work appears to be carefully done in line with previous excellent publications by Carroni, the main findings described in the current manuscript are not particularly new or exciting. Reports from the Sauer group and more recently from the Goldberg/ Rubin have already demonstrated this kind of allosteric regulation between the 2 partners (protease and unfoldase). See for example (just for ClpC) Li et al "Structure and Functional Properties of the Active Form of the Proteolytic Complex, ClpP1P2, from Mycobacterium tuberculosis" or " Substrate delivery by the AAA+ ClpX and ClpC1 unfoldases activates the mycobacterial ClpP1P2 peptidase" by the Sauer group. In the first it is shown for example that binding of ClpC to ClpP increases the ClpC1 ATPase activity – to note that the authors here specifically discuss ATPase activation in the introduction ignoring this was already tested. Maybe the authors do not know these manuscripts, as they are not cited or referred, but the reciprocal activation between unfoldase (ClpC) and ClpP is not new and it is indeed already well established in the field. This said, we felt it is surely not appropriate to use the current title, that claims somehow the credit of this finding, and we believe that other title more focused on the structure determination is appropriate. As the authors know the present manuscript concerns mostly structural biology and in its core mostly supports previous observations.

We thank the reviewer for their valuable comments. We have changed the title of the manuscript to "**Structure of the central *Staphylococcus aureus* AAA+ protease MecA/ClpC/ClpP**". We have revised the introduction and the discussion according to the advice and referred to the indicated published work as well as to the most recent work on the subject that was published or put in pre-print lately.

*1)As stated above the following articles should be read and cited
"Substrate delivery by the AAA+ ClpX and ClpC1 unfoldases activates the mycobacterial ClpP1P2 peptidase"*

This work (Schmitz and Sauer 2014) is now cited and discussed as it shows the allosteric stabilizing effect of the AAA+ partner over the ClpP1P2 peptidase in *Mycobacterium*.

“Structure and Functional Properties of the Active Form of the Proteolytic Complex, ClpP1P2, from Mycobacterium tuberculosis”

This work (M. Li et al. 2016) is now cited and discussed as it shows the allosteric activating effect of the Bz-LL-activated ClpP1P2 over the ClpC1 ATPase in *Mycobacterium*. This is in good agreement with the *S. aureus* case presented in our study.

“Communication between ClpX and ClpP during substrate processing and degradation” likely the first article showing the reciprocal activation and modulation between the partners.

This work (Joshi et al. 2004) is now cited and discussed. It shows that in *E. coli* the peptidase ClpP (WT) reduces the ATPase and unfolding activity of ClpX-WT, while it rescues ATPase defects in ClpX mutants at the same time. Notably, different observations have been reported for *Mycobacterium tuberculosis*, where ClpP1P2 association enhances ClpX ATPase activity (Li et al. 2016). The mechanistic reasons underlying the differing effects of peptidase association on ClpX ATPase activity are unclear.

2) The authors state that ClpP association reduces ATPase activity in ClpX in the introduction but previous results with MtbClpC (that does associate with MecA but is more similar than ClpX to SaClpC) have shown already that an ATPase activation is observed (see “Structure and Functional Properties of the Active Form of the Proteolytic Complex, ClpP1P2, from Mycobacterium tuberculosis”) this should be cited and discussed.

We have added the respective work (Li et al. 2016) to the manuscript. These findings pinpoint similarities in the cooperations between Sa ClpC and Mtb ClpC1 and their respective ClpP and ClpP1/ClpP2 peptidases and underline the role of the protease partner in AAA+ protein activation.

3) We are aware that writing structural biology is a hard task but an effort should be made to make the article more approachable.

This is the first structure of a ClpC-ClpP complex – excluding the things we already know from other AAA+ proteins and are conserved in ClpC - what are the major differences observed (if any)? What is really new? We would suggest some rewriting to emphasize what is really new and make the manuscript less descriptive and therefore lighter for the reader.

As we have mostly re-written the manuscript adding new data, we hope to have managed making the reading smoother. Figures are also mostly new and hopefully they convey the main pieces of information that we can gather from our data: (i) an analysis of the MecA part of the complex and its motion (new Fig. 2) and (ii) a possible rational for the activation of the ClpC ATPase activity via ClpP (new Fig.6).

For example: The authors test several beta hairpin ClpP mutants for peptidase activity (some are inactive) to later describe that several of the same mutants fail to form the 14 subunit ClpP – not surprising they also fail to function together with ClpC. As we introduce mutations in proteins folding or major structural modifications are not unexpected – it would be more simple to just state that these mutants fail to form the ClpP complex instead of testing them in other assays and show that they not work – what conclusions were the author expecting to obtain from these mutants if they do not form the ClpP barrel?

We followed the suggestion of the reviewer and shortened repetitive parts of the manuscript and data discussion of the ClpP mutants that exhibit oligomerization defects.

Reviewer #2

1. *I would recommend that the authors change the title of the manuscript as it is not very informative. They should mention in the title that they have a MecA-ClpC-ClpP structure. The issue of reciprocal allosteric regulation is already known for ClpP-AAA+ complexes.*

We agree with the reviewer and it was also pointed out by reviewer #1. We have changed the title to: Structure of the central *Staphylococcus aureus* AAA+ protease MecA/ClpC/ClpP”.

2. *More details are needed for the cryoEM data analysis. The data processing scheme needs to be shown. Also, did the authors only identify one state of the MecA/ClpC/ClpP complex?*

We agree with the reviewer and for this completely new dataset and data analysis we have reported the processing scheme (new Supplementary Fig. 1). Same as for the old dataset, we only observe one state of the MecA/ClpC/ClpP complex even upon extensive classification, both global and focused. We have also a ClpC double Walker B (ATPase deficient mutant) dataset (in complex with MecA and ClpP) that also only shows one state. This is different from most other AAA+/ClpP complex structures determined by cryo-EM to date. We reason that this could be caused by the presence of the MecA crown that partially restrains the ClpC movements or makes them more synchronised. We are depositing the raw data in EMPIAR so that they will be available for anyone testing new analysis method and dig possible rare conformations.

3. *Figure 2a, label the subunits of ClpP.*

The new manuscript contains completely new Figures. Hopefully, labelling is better and more consistent. The labelling (subunits a-f for ClpC and Pa-Pn for the ClpP protomers) match the chain ID of the deposited pdb models.

4. *Page 10 – The use of ADEP1 to propose that the ClpC P-loops undergo rapid cycling is not founded. What the experiments demonstrate is that binding of ADEP1 to ClpP can out compete the binding of ClpC to ClpP. How fast ClpC cycles in its binding to ClpP is not relevant in these experiments. To make a clearer conclusion, the authors need to measure Kd's and kon and koff rates. The authors should reword this section or do more experiments.*

We agree with the concern of the reviewer and have removed the respective data from the manuscript.

5. *Provide a better view in the respective Figure for this description: “The ClpP N-terminal b-hairpins contact the ClpC inner channel more on one side of the ring (ClpC protomers f-c (Fig. 2a/e).”*

The new manuscript contains a better refined map and completely new figures. We can now show the specific contacts of each ClpC subunits to each ClpP protomer β -hairpin (new Fig. 5e).

6. *Given that the binding of SaClpP to SaClpC enhances the ATPase activity of SaClpC, while the opposite is found for the binding of EcClpP to EcClpX, it would be important*

for the authors to carry out a detailed analysis of these observations using their cryoEM structure of SaClpCP-MecA and the published cryoEM structures of EcClpXP.

The inhibitory effects of *E.coli* ClpP on ClpX ATPase activity is specific as activation of *M. tuberculosis* ClpX ATPase activity upon association with ClpP1P2 has been described, as also pointed out by reviewer 1 (see above) (M. Li et al. 2016). Furthermore, *E. coli* ClpP can increase ATPase activity of ClpX mutants (Joshi et al. 2004). Thus, the impact of *E. coli* ClpP on ClpX activity is context dependent. The available data therefore do not support major mechanistic differences in the cooperation between ClpC/ClpX with ClpP. Accordingly, the contact sites between *Neisseria meningitidis* ClpX (Ripstein et al. 2020) and ClpP and the ones described in this study are similar, including the interaction between ClpP β -hairpins and the allo-W region of the AAA+ partner. We added a respective statement to the manuscript.

As it had been also required by reviewer #3, we have looked at the published *E. coli* ClpX/ClpP cryo-EM structures (Fei et al. 2020) in the regions of interaction (allo-W, allo-R and P-loops) at the interface between a AAA-2 subunit with ADP and one with ATP bound (see Figure 1 below). The overall pattern of interactions, between the β -hairpins and the P-loops and between the β -hairpins and the allo-R region preceding the R-finger, are maintained. However, there are major differences between *E. coli* ClpX and *S. aureus* ClpC, as expected by the low sequence identity (see alignment below). The P-loop in ClpX is longer, but, as in ClpC, it does get contacted by the ClpP β -hairpin in the ATP bound subunit. The allo-R region is made of three helices instead of two, but, as in ClpC, it does get contacted by ClpP β -hairpin in the ADP-bound subunit. The allo-W residue in *E.coli* ClpX (D187) is not in contact with the ClpP β -hairpins. In *N. meningitidis* ClpX/ClpP the ClpP β -hairpin has been reported to contact the ClpX pore-loop 2 that correspond to the allo-W. It is thus very difficult to draw any conclusion about specific mechanisms of activation or reduction of the ATPase activity of ClpX by ClpP. It should also be noted that the overall conservation between ClpC and ClpX in the ClpP-interacting regions is not high (see Supplementary Fig. 5a and Figure 2), further complicating the direct comparison of ClpX vs ClpC cooperations with ClpP. We therefore prefer not adding such comparison to the manuscript. We believe this is also outside the scope of this study.

E.coli ClpX/ClpP pdb code 6p01, subunit E-ATP and F-ADP:

Figure 1 Visualisation and analysis of the interface between an ADP and and ATP bound subunit in *E.coli* ClpX from the ClpX/ClpP complex (pdb 6p01, (Fei et al. 2020)). This interface and contacts are the same in *N. meningitidis* ClpX/ClpP configurationA (Ripstein et al. 2020). Overall the pattern of interactions between the ClpP β -hairpins and the ClpX regions adjacent to the ATP pockets (allo-R in magenta) and to the ClpX P-loops (P-loop in green) are similar to the *S.aureus* ClpC/ClpP. However, compared to *S.aureus* ClpC/ClpP where two adjacent ClpP β -hairpins contact two adjacent ClpC subunits, in *E.coli* ClpX there is in the middle a ClpP β -hairpin that doesn't contact any ClpX subunit.

Figure 2: **Sequence alignment** of the ClpP-interacting AAA domain of ClpC, ClpA and ClpX from *Staphylococcus aureus*, *Bacillus subtilis* and *Escherichia coli*. Postions of Walker A and B motifs (red) and ClpP-interacting parts (pore-2 loop/allo-W, P-loop, allo-R) of ClpC, ClpA and ClpX are indicated.

A comparison to ClpA is provided in the answer to reviewer #3 below.

Reviewer #3

1. *The manuscript would significantly be improved by providing a detailed assessment of how mutations may relate to the structure. Given that the authors have obtained an atomic model of the ClpP-ClpC interactions, they should be able to relate the mutagenesis results to the structural interactions, and provide insights into how mutations would disrupt complex formation and function. In particular, while the authors showed that ClpP enhances the activities of its AAA+ partners, the authors don't propose a mechanism by which this modulation of the activities. Further insights into an allosteric regulatory function could be gained through mutations in regions in alloR and alloW that contact the beta-hairpins. Overall, the authors state what they observe without providing mechanistic relevance that is tied to the functional and biochemical studies.*

Using the new structure, we can now describe more accurately the contacts between the β -hairpins from different ClpP protomers and the ClpC subunits (new Fig. 5). This also allows us to explain phenotypes of ClpP β -hairpin mutants. We additionally propose a possible mechanism, described in the revised manuscript and in the new Fig. 6, which implies a readjustment of the ATP pocket and of the R-finger at the interface between a AAA2 domain that has undergone hydrolysis and its neighbour,

which could/should hydrolyse next. This repositioning of the ATPase pockets is facilitated by contacts mediated by the allo-W and allo-R regions. So, we followed the suggestion of the reviewer and studied the consequences of ClpC alloW (E620A/K621A) and alloR (E692A/K694A) mutants on activation by ClpP. Additionally, we analyzed the role of the conserved N659 residue, which our structural analysis implicates in signaling between ClpP docking sites and the AAA2 ATPase center. We found that the alloW and N659A mutants have reduced degradation activity (new Fig.7 and Supplementary Fig.10), which could be linked to reduced ATPase activity. This can be rationalized by close vicinity of the alloW region and N659 to the Walker B motif and bound nucleotide and thus the catalytic site of the AAA2 domain. Importantly, ClpC-E620A/K621A and ClpC-N659A ATPase activities were no longer stimulated in presence of ClpP (new Fig. 7e). Similarly, unfolding activities of ClpC-E620A/K621A and ClpC-N659A were much less enhanced upon ClpP association as compared to ClpC-WT (new Fig. 7f, Supplementary Fig.10d-e). In contrast, the alloR mutant E692A/K694A showed WT-like degradation activities. Notably, this mutant showed a partially reduced ATPase activity that was restored to WT-like activities upon ClpP association. This suggests that ClpP binding compensates for the partial defects caused by the allo-R mutation. We conclude that ClpP contacts with the allo-W region are crucial for ATPase activation. Contacts between ClpP and allo-R are instead more complex and point on a potential role of fine tuning ClpC-ClpP cooperation.

2. *One finding that warrants further investigation is the ADEP1-mediated restoration of peptidase activity in mutant ClpP. This is particularly interesting, given that the majority of these mutants don't form a functional oligomer. Is ADEP1 mediating inter-ClpP interactions to enable oligomer formation? How could such a mechanism function? Structural insights into the ADEP1 restoration of ClpP activity would vastly increase the impact of this finding.*

ClpP represents a highly allosteric system, in which binding of e.g. ADEP1 induces long-range conformational changes including the N-terminal β -hairpins and interactions between ClpP monomers and heptameric rings. Similarly, mutations in the N-terminal loops have been shown to affect structural integrity of the *S. aureus* ClpP barrel, underlining allosteric communications between diverse ClpP sites (3). Structural defects of *S. aureus* N-terminal loop mutants can be rescued by ADEP binding (Vahidi et al. 2018). We consider this observation very similar to ours: the partial rescue of N-terminal loop mutants upon ADEP binding. ADEP binding leads to ordering of N-terminal loops into β -hairpin structures (D. H. S. Li et al. 2010) and this effect likely restores proper β -hairpin formation in our β -hairpin mutants. Notably, ADEP binding is crucial for oligomerization of *B. subtilis* and human ClpP (Kirstein et al. 2009; Lowth et al. 2012), underlining the impact of the compound on ClpP ring formation and stability. We have added these clarifications to the revised manuscript.

3. *While there are numerous comparisons to other ClpP-related systems throughout the manuscript, the authors don't detail these comparisons in accompanying figures – this would be particularly useful for structural details that the authors identify to be unique to ClpP-ClpC.*

We have partially answered this question to reviewer #2, regarding comparison to *E.coli* ClpX/ClpP (Fei et al. 2020). In this new amply revised manuscript, we have a more complete analysis of each ClpC subunit contact with each ClpP protomer β -hairpin and we refer to the relevant similarities for other AAA+/ClpP systems. The overall pattern of interaction and the residues that we have identified as important in the allo-W and allo-R regions are conserved in the Hsp100 proteins ClpC, ClpA and ClpE (new Supplementary Fig. 5a and Figure 2 above). We do not endeavour in a comparison of each and every subunit of all the existing Hsp100s in complex with ClpP. Such a detailed analysis could be the subject of a review, but ideally it

should be supported by mutational analysis in the different AAA+ protease systems. Such mutational analysis is provided here for ClpC/ClpP but is missing for the other AAA+ proteases.

We also looked at the *E. coli* ClpA/ClpP structures for the subunits at the interface between the inactive ADP-bound part of the ring and the ATP-bound active adjacent subunit. The positions of the allosteric regions (allo-W, allo-R and P-loops) are conserved in ClpA relative to the ClpP protomer β -hairpins underneath (see Figure 3 below). When we overlap the subunits at the inactive/active interfaces from different configurations of the ClpA/ClpP (Lopez et al. 2020) with the one from *S. aureus* ClpC/ClpP we see an almost perfect overlap of all domains, but a change in the relative positions of the other AAA+ subunits and more interestingly of the ClpP barrel. The mechanistic relevance of these structural differences is, however, unclear as *E. coli* ClpP association also enhances ATPase activity of ClpA. We therefore prefer not adding such comparison to the manuscript. We also believe that such analysis will be outside the scope of this study.

Figure 3 Superimposition of two subunits making the interface between the inactive, ADP-bound, and active ATP-bound part of the AAA-2 ring in the *S.aureus* ClpC/ClpP complex (this study) and *E.coli* ClpA/ClpP complex. The high sequence and structural conservation of the P-loop, allo-R and allo-W regions suggest that the allosteric effect of ClpP onto the AAA+ partner and its mechanism can be conserved in Hsp100 unfoldases.

- In the discussion, authors propose a “sensing” function, as opposed to a direct stimulating effect of ClpP beta-hairpin on ATP hydrolysis. This proposal seems incongruent with the findings that ClpP binding to ClpC stimulates its ATPase activities. This should be reconsidered or the proposed mechanism further clarified in light of the findings.*

As explained in the answer to the first question above, we now propose a possible way of ClpP facilitating sequential and more efficient ATP hydrolysis around the ClpC ring. The model is partially validated by ClpC mutants (e.g. allo-W E620A/K621A), which are no longer stimulated by ClpP association (new Fig 7e). Since the allo-R mutant E692A/K694A is still activated by ClpP, we speculate that the allosteric activation of ClpP onto ClpC is far more intricate and tunable by a number of residues, than the simple activation path that we propose (new Fig. 6). In an attempt to propose a detailed mechanism, we started molecular dynamics simulations of the complex. As these calculations are fairly expensive, we plan to release the results in an upcoming publication.

5. *The introduction is a bit long-winded covering aspects that are not relevant to the major findings of the paper. This could be condensed while still being approachable to non-experts in ClpP-ClpC interactions. The discussion section mostly reiterates findings already discussed in the manuscripts, as opposed to a discussion that integrates the findings into a cohesive allosteric mechanism that constitutes an advance in our understanding of this important complex.*

We followed the suggestion of the reviewer and completely revised introduction and discussion accordingly also in light of new results published by other groups in the meantime.

6. *-The quality of the EM density for the nucleotides is difficult to assess, as the authors have "zoned" the cryoEM map to only display voxels within a certain radius of the nucleotide to show that EM density is present for the model. Low contour values can be used to produce density surrounding any modeled atom within a map. While I do not suspect that the authors are being intentionally disingenuous in their analysis, the density for surrounding residues (particularly Arginine fingers) should be included in this figure or in a supplementary figure depicting this region. Based on Figure 4a, the density for the nucleotides doesn't seem of sufficient quality to unambiguously assign nucleotide state.*

In this completely revised manuscript, we present new structural data and we show all the density for the nucleotides and surrounding areas (new Fig. 6 a). We also report in the text and caption about any doubtful interpretation of the density, such as in the case of the ATP pocket of subunit a AAA2. Built models and maps (both sharpened and unsharpened) are available for the reviewers in a shared folder provided to the editor (link).

7. *Importantly, the EM methods and validation is lacking in the manuscript, which makes overall assessment of the quality of the models challenging.*

We have addressed all points raised by the reviewer and added respective information to the methods section and in the statistics table. For completeness we also make the models and maps available and we are depositing the raw data to EMPIAR.

References

- Fei, Xue, Tristan A. Bell, Simon Jenni, Benjamin M. Stinson, Tania A. Baker, Stephen C. Harrison, and Robert T. Sauer. 2020. "Structures of the ATP-Fueled ClpXP Proteolytic Machine Bound to Protein Substrate." *ELife* 9 (February). <https://doi.org/10.7554/eLife.52774>.
- Joshi, Shilpa A., Greg L. Hersch, Tania A. Baker, and Robert T. Sauer. 2004. "Communication between ClpX and ClpP during Substrate Processing and Degradation." *Nature Structural & Molecular Biology* 11 (5): 404–11.
- Kirstein, Janine, Anja Hoffmann, Hauke Lilie, Ronny Schmidt, Helga Rübsamen-Waigmann, Heike Brötz-Oesterhelt, Axel Mogk, and Kürşad Turgay. 2009. "The Antibiotic ADEP Reprogrammes ClpP, Switching It from a Regulated to an Uncontrolled Protease." *EMBO Molecular Medicine* 1 (1): 37–49.
- Li, Dominic Him Shun, Yu Seon Chung, Melanie Gloyd, Ebenezer Joseph, Rodolfo Ghirlando, Gerard D. Wright, Yi-Qiang Cheng, Michael R. Maurizi, Alba Guarné, and Joaquin Ortega. 2010. "Acyldepsipeptide Antibiotics Induce the Formation of a Structured Axial Channel in ClpP: A Model for the ClpX/ClpA-Bound State of ClpP." *Chemistry & Biology* 17 (9): 959–69.
- Li, Mi, Olga Kandror, Tatos Akopian, Poorva Dharkar, Alexander Wlodawer, Michael R. Maurizi, and Alfred L. Goldberg. 2016. "Structure and Functional Properties of the Active Form of the Proteolytic Complex, ClpP1P2, from Mycobacterium Tuberculosis." *The Journal of Biological Chemistry* 291 (14): 7465–76.

- Lopez, Kyle E., Alexandra N. Rizo, Eric Tse, Jiabei Lin, Nathaniel W. Scull, Aye C. Thwin, Aaron L. Lucius, James Shorter, and Daniel R. Southworth. 2020. "Conformational Plasticity of the ClpAP AAA+ Protease Couples Protein Unfolding and Proteolysis." *Nature Structural & Molecular Biology* 27 (5): 406–16.
- Lowth, Bradley R., Janine Kirstein-Miles, Tamanna Saiyed, Heike Brötz-Oesterhelt, Richard I. Morimoto, Kaye N. Truscott, and David A. Dougan. 2012. "Substrate Recognition and Processing by a Walker B Mutant of the Human Mitochondrial AAA+ Protein CLPX." *Journal of Structural Biology* 179 (2): 193–201.
- Ripstein, Zev A., Siavash Vahidi, Walid A. Houry, John L. Rubinstein, and Lewis E. Kay. 2020. "A Processive Rotary Mechanism Couples Substrate Unfolding and Proteolysis in the ClpXP Degradation Machinery." *ELife* 9 (January). <https://doi.org/10.7554/eLife.52158>.
- Schmitz, Karl R., and Robert T. Sauer. 2014. "Substrate Delivery by the AAA+ ClpX and ClpC1 Unfoldases Activates the Mycobacterial ClpP1P2 Peptidase: Substrate Activation of Mycobacterial ClpP." *Molecular Microbiology* 93 (4): 617–28.
- Vahidi, Siavash, Zev A. Ripstein, Massimiliano Bonomi, Tairan Yuwen, Mark F. Mabanglo, Jordan B. Juravsky, Kamran Rizzolo, et al. 2018. "Reversible Inhibition of the ClpP Protease via an N-Terminal Conformational Switch." *Proceedings of the National Academy of Sciences of the United States of America* 115 (28): E6447–56.

Answer to reviewers

For the second round of revision of the manuscript "Structure of the central Staphylococcus aureus AAA+ protease MecA/ClpC/ClpP" by Azinas et al.

Comments of the reviewers are shown in blue.

We thank the reviewers for their positive evaluation of our work and the time they spent on it. We have addressed the concerns of reviewer #1 in the final version of the manuscript. Detailed answers are given below.

Reviewers' comments:

Reviewer #1 (Remarks to the Author):

The authors have clearly made a considerable effort to improve the manuscript, which I view positively. They have also addressed comments from the various reviewers.

I had no major concerns regarding the structural data. The structures of AAA+ unfoldases in complex with their partner proteases are inherently challenging to obtain, and in my view, the quality of the structural data alone would justify the publication of this work. In this revised version, the authors further improve the structural component and also attempt to expand the associated biochemical analysis. Again, I find no issue with the structural work — as is often the case from the Carroni group, it appears to be carefully and thoroughly executed.

Thanks!

However, I find some of the new biochemical data, particularly the mixing experiments presented in Fig. S8, difficult to interpret. I do not understand what the authors intend to demonstrate with these assays. First, the term "concentration-sensitive range" is unclear and should be defined more precisely. Additionally, what conclusions can truly be drawn from mixing WT ClpC with mutant ClpC? Do they form mixed hexamers? Are the association constants for WT and mutant ClpC the same? None of these foundational assumptions are addressed or experimentally demonstrated. Furthermore, the use of buffer as a control is not ideal. A better control would be to use an unrelated protein at similar concentration, or to test the effect of adding increasing amounts of WT protein and observing whether ATPase activity increases proportionally.

In addition, the statement that 4–5 subunits need to be active for ClpC function is not convincingly supported by the current data. To rigorously test this, the authors would need to adopt an approach like that used by the Sauer group for ClpX, involving covalently linked hexamers. This allows specific subunits to be mutated while keeping the hexamer intact, making it possible to test how many inactive subunits are compatible with activity. I understand that constructing covalently linked ClpC hexamers is technically challenging — and may not even yield a soluble or functional complex — but in light of this, I would recommend removing the current mixing experiments from the manuscript, as they are not informative and may detract from the overall clarity.

We agree with the reviewer that it is hard to proof the stoichiometry of the poisoning effect of inactive ClpC subunits in the hexameric complex using indirect experiments. The chimeric fusion approach used for ClpX is hardly feasible for ClpC because of its large size and that is why we opted for the mixing experiments. We have decided to remove these results from the manuscript, as suggested by the reviewer.

Lastly, I want to reiterate a point from my previous review: the continued use of ClpP

mutants that fail to assemble properly is problematic. While I appreciate the effort required to generate and purify these mutants, if a ClpP variant does not form a complex, any downstream functional assays involving ClpC will be inherently compromised and hard to discuss. Including data from such mutants across multiple assays does not add value and only makes the manuscript harder to read and interpret. Since this issue has already been noted, and no new experiments are needed, I would suggest simply removing or minimising the use of those non-assembling mutants to improve clarity - just focus on the other ones.

We have followed the suggestions of the reviewers and removed the data and discussion of the ClpP mutants defective in complex formation from the manuscript. Only in SDS-gels (Suppl. Figures S8a, S9c) we have left the lines of the mutants defective in complex formation to avoid cropping the gels. The oligomerization deficient ClpP mutants are labeled in those gels.

Reviewer #2 (Remarks to the Author):

The authors have addressed my comments as best as they can. I do not have further comments.

Thanks!

Reviewer #3 (Remarks to the Author):

I appreciate the efforts the authors put into addressing our concerns, and believe that the manuscript is much improved. I am satisfied with the responses and find the manuscript appropriate for publication.

Thanks!